



# Sensitivity of precipitation and temperature over Mount Kenya area to physics parameterization options in a high-resolution model simulation performed with WRFV3.8.1

Martina Messmer[1,2,3], Santos J. González-Rojí[1,2], Christoph C. Raible[1,2], and Thomas F. Stocker[1,2]

[1]Climate and Environmental Physics, University of Bern, Bern, Switzerland
[2]Oeschger Centre for Climate Change Research, University of Bern, Bern, Switzerland
[3]School of Earth Sciences, The University of Melbourne, Melbourne, Victoria, Australia

**Correspondence:** Martina Messmer (martina.messmer@climate.unibe.ch)

**Abstract.** Several sensitivity experiments with the Weather Research and Forecasting (WRF) model version 3.8.1 have been performed to find the optimal parameterization setup for precipitation amounts and patterns around Mount Kenya at a convection-permitting scale of 1 km. Hereby, the focus is on the cumulus scheme, with tests of the Kain-Fritsch, the Grell-Freitas and no cumulus parameterization for the parent and all nested domains. Besides, two long wave radiation schemes and two planetary boundary layer parameterizations are evaluated. Additionally, different nesting ratios and numbers of nests are tested. The precipitation amounts and patterns are compared against a large number of weather station data and three gridded observational data sets. The temporal correlation of monthly precipitation sums show that fewer nests lead to a more constrained simulation and hence, the correlation is higher. The pattern correlation with weather station data confirms this result, but when comparing it to the most recent gridded observational data set the difference between the number of nests and nesting ratios are marginal. The precipitation patterns further reveal that the Grell-Freitas cumulus parameterization provides the best results, when it comes to precipitation patterns and amounts. If no cumulus parameterization is used, the temporal correlation between gridded and in-situ observations and simulated precipitation is especially poor with more nests. Moreover, even if the patterns are captured quite well, a clear overestimation in the precipitation amounts is observed around Mount Kenya when using no cumulus scheme at all. The Grell-Freitas cumulus parameterization also provides reasonable results for 2-metre temperature with respect to gridded observational and weather station data.

## 1 Introduction

East Africa, including Kenya, has anomalously dry climate conditions compared to many other equatorial regions around the globe (e.g., Trewartha, 1981; Nicholson, 2017). The precipitation patterns in East Africa are very heterogeneous, which can be attributed to the variety and complexity of large-scale controls, i.e., topography, influence from the ocean, the dynamics of





the tropical circulation and lakes (Nicholson, 2017). The topography, in particular the Turkana channel between the Ethiopian and East African highlands in Kenya, exerts a strong steering effect on the low-level flow on time scales from seasons to days (Paegle and Geisler, 1986; Slingo et al., 2005). The Turkana jet has an influence on the local climate and especially on precipitation, and a study by Nicholson (2016a) suggests that it might even be responsible for the suppression of the summer

rainy season in northwestern Kenya. The zonal circulation over the Indian Ocean further influences precipitation in Kenya, as it is located under subsiding air masses, leading to the aforementioned aridity over an equatorial region (Pohl and Camberlin, 2011; Nicholson, 2017). This circulation and thus, the intensity and the vertical extent of the subsidence account for variations in the inter-annual rainfall variability over Kenya (Pohl and Camberlin, 2011; Nicholson, 2017). The second largest freshwater lake in the world, Lake Victoria, also contributes to rainfall in this area. It generates its own mesoscale atmospheric circulation

system that leads to high rainfall amounts over the lake, where lake surface temperatures are strongly related to the rainfall amounts (Sun et al., 2014). Furthermore, local thunderstorms with heavy precipitation can be triggered over Lake Victoria, rendering the lake-land breeze but also large-scale moisture availability as the main control (Thiery et al., 2015; Woodhams et al., 2019).

All these large-scale controls lead to the fact that the climate in Kenya is characterized by two rainy seasons. The March-

April-May (MAM) season is often termed as 'long rains', as this season is associated with the longest lasting and heaviest precipitation events. The other rainy season is called 'short rains' and occurs in October-November (ON). It plays a less important role in the total amount of precipitation, but accounts for most of the inter-annual variability (Camberlin and Philippon, 2002; Hastenrath et al., 2010). Thus, it is not surprising that the 'short rains' are responsible for both floodings and droughts. The occurrence of floods in Kenya is not unusual, and often floods set in after very dry years (Parry et al., 2012). Droughts

are found to be related to El Niño Southern Oscillation (ENSO) events on inter-annual timescales, as it affects the atmospheric circulation over the Indian Ocean. This circulation has also an impact on the 'short rains' in East Africa (Pohl and Camberlin, 2011; Nicholson, 2016b). Additionally, the Madden-Julian-Oscillation can impact precipitation on inter-seasonal timescales and it is able to strengthen or weaken the climatological convective and dynamic zonal gradients between Southeast Asia and East Africa (Pohl and Camberlin, 2011). The low-level jet stream in the Turkana channel is also suggested to be able to enhance

extremes in precipitation over East Africa (Nicholson, 2016b). Nevertheless, in the recent past, droughts instead of floods are of major concern in Kenya. The more frequent occurrence of droughts seems to be related to an ongoing downward trend in the 'long rains' in MAM that has commenced in the 1980s and lasted up to the late 2000s (Williams and Funk, 2011; Liebmann et al., 2014; Ayugi et al., 2016).

The rather sparse observation network in East Africa and also Kenya in combination with the aforementioned complexity

of the climate, conspire against obtaining a better understanding of all the involved processes that dominate the climate, but also its changes. To overcome this issue, climate models, and in particular regional climate models, could help understanding those processes in more detail. Nevertheless, capturing the convective precipitation in the tropics correctly is also a challenge for regional climate models, and that is why several studies focus on the evaluation of their performance in different regions (e.g., Rauscher et al., 2010; Kendon et al., 2017; Brune et al., 2020; Wu et al., 2020).





Only in the past few years the number of simulations over Africa or over Eastern Africa has increased. At the same time, the resolution of these simulations has become much finer. Cook and Vizy (2013) performed a simulation over entire Africa using the Weather Research and Forecasting (WRF) Model (Skamarock et al., 2008) at a 90 km horizontal resolution. They concluded that the model is able to capture the distribution of the precipitation and the corresponding circulation quite well over Eastern Africa, but a wet bias in the model simulation remains. Williams et al. (2015) found an overestimation of precipitation

and a well captured spatial pattern over the Lake Victoria basin, which is inline with the results found in Cook and Vizy (2013). Williams et al. (2015) used the UK Met Office Hadley Centre Regional Climate Model at 50 km horizontal resolution over Africa. Two simulations, one with 50 and the other with 25 km resolution over Eastern Africa were also performed by Kerandi et al. (2017). They examined the representation of temperature and precipitation over the Tana river basin in Kenya, finding that temperature and precipitation patterns are well captured, but with a cold temperature bias. The increase in the resolution from 50

to 25 km resulted in a much better representation of precipitation (Kerandi et al., 2017). Otieno et al. (2019) performed different sensitivity studies with WRF to test the effect of four cumulus parameterizations (Kain-Fritsch, Kain–Fritsch with a moisture advection-based trigger function, Gréll-Dévényi, and Betts–Miller–Janjicon schemes) to the representation of precipitation over East Africa during wet years. The authors still used a rather coarse resolution of 36 km covering East Africa including parts of the Indian Ocean and the rain forest in Congo, i.e., two important moisture sources.

The most recent simulations over (Eastern) Africa access the convection-permitting scales (resolution finer than 5 km) (Stratton et al., 2018). This scale can have a fundamental impact on model variables, in particular on precipitation (Ban et al., 2014; Giorgi et al., 2016; Gómez-Navarro et al., 2018). This is especially true for regions with high and complex topography such as East Africa. The simulation published in Stratton et al. (2018) is performed with the Met Office Unified Model at 4.5 km horizontal resolution. Due to its high spatial resolution, it is convection-permitting and hence, a cumulus parameterization is

not needed. This simulation is investigated in more detail in Finney et al. (2019) with respect to East African climate and compared to a parameterized 25 km spatial resolution simulation. They find that especially the diurnal cycle in rainfall benefits from the convection-permitting resolution, but also precipitation intensities and patterns improve. Additionally, Woodhams et al. (2018) confirmed that a convection-permitting simulation is able to better represent the sub-daily precipitation intensities over Lake Victoria and the occurrence of storms over land.

The studies presented above already indicate that there are several regional climate models (RCMs) available. Each of these models have different sets of parameterizations that can be chosen, and the ability to simulate the climate over a certain region depends to a large part on the selection of the different parameterization options. Several studies have evaluated the transferable skills of RCMs to different regions (e.g., Takle et al., 2007; Jacob et al., 2007; Rockel and Geyer, 2008; Jacob et al., 2012; Bellprat et al., 2016; Russo et al., 2019). However, a clear answer to the transferability has not been found yet. This is why

RCMs still need to be retuned for different regions, as every region has its particular climate with specific drivers, which are not equally well captured by different parameterization options. Hence, this study presents a set of sensitivity studies performed by WRF, initiated and driven by ERA5, to find an optimal setting for the representation of precipitation in convection-permitting simulations over Mount Kenya.





In this paper, the focus on Mount Kenya is chosen, as it plays a crucial role in the supply of freshwater both in the highlands and in the surrounding lowlands (Liniger et al., 2005). The availability of fresh water decreases drastically with longer distances from Mount Kenya, and is further reduced by evapotranspiration from the vegetation in the drier savannas of the lowlands (Ngigi et al., 2007). Population growth through migration puts further pressure on water availability (Ngigi et al., 2007) which may result in disputes, marginalization and conflicts (Wiesmann et al., 2000). This situation is exacerbated by progressive climate change that will affect water availability through changes in precipitation amounts and patterns, induced by either local or large scale changes. To understand the behaviour of precipitation in this complex topographical area and to obtain possible adaptation strategies, it is vital to create reliable regional climate simulations that can also be used for climate projections in a next step.

The paper gives a detailed description of the sensitivity simulations performed with WRF, but also its initial and boundary conditions provided by the reanalysis data ERA5. Furthermore, the different observation based gridded data for precipitation and temperature and the weather station data are presented in Section 2. Section 3 provides an analysis of the temporal and spatial representation of precipitation patterns over the area around Mount Kenya. Also, the sensitivity of the different parameterization options to precipitation amounts and patterns are investigated. The analysis is topped off with a brief description of the 2-metre temperature around Mount Kenya. Finally, the paper is wrapped up by summarizing and concluding remarks in Section 4.

## 2 Model configuration and Data

### 2.1 WRF Model

We adopt the regional climate model WRF (version 3.8.1; Skamarock et al., 2008) to obtain fine scale and local precipitation patterns. This model allows us to dynamically downscale initial and boundary conditions, which in this study are provided by ERA5 reanalysis. To determine an optimal setup for Kenya, and particularly the Mount Kenya area, we test different parameterization schemes, focusing on cumulus parameterizations, with two different model setups and nesting ratios. The experiments are described in more detail in the following and are summarized in Table 1. The experiments are all run for the same period of time, i.e., the year 2008. To permit the soil and the atmosphere to adjust to the initial conditions, we allow for two month of spin-up. Since the soil variables are well equilibrated in the ERA5 data, the used spin-up time of two month in our simulations should be enough to bring the soil and the atmosphere into an equilibrium. Previous studies (Angevine et al., 2014; Jerez et al., 2020; Velasquez et al.) back-up the idea that rather short spin-up periods are enough for variables such as temperature or precipitation to reach the equilibrium in WRF, but longer periods are recommended (a few months). This means that the simulations start on the 1$^{st}$ of November 2007 and end on the 31$^{st}$ of December 2008.

Two different nesting ratios, i.e., 1:3 and 1:5, have been used in different model domain settings, to estimate the effect of the nesting ratio on the modelled precipitation and temperature. For the nesting ratio of 1:3, a four domain (i.e., 27, 9, 3, 1 km horizontal resolution) and a three domain (i.e., 9, 3, 1 km horizontal resolution) setup have been tested. Also the 1:5 nesting ratio is run with two different setups, i.e., a three nested (25, 5, 1 km horizontal resolution) and a two nested domain setup (5,





1 km horizontal resolution). To test if the coarser setups affect the representation of precipitation and temperature over the study area, simulations with a coarser and finer parent grid are performed. This is because for the coarser setups the downscaling resolution is very similar to the one of ERA5, which provides the initial and boundary conditions. Note that in the simulations

with a reduced number of nests (3 domains instead of 4 for the 1:3 ratio, and 2 instead of 3 for the 1:5 ratio), the parent domain always corresponds to the second domain of the experiment with one nest more (Table 1). All simulations have 49 vertical eta levels up to 50 hPa and an innermost domain located over Mount Kenya with 1 km horizontal resolution. When comparing the different sensitivity experiments in the results (section 3), the focus is always on the innermost domain of all the simulations. This is, because all the simulations resolve this domain with 1 km horizontal grid spacing. To save some computational costs,

an adaptive time step is used, which is between 54 and 810 seconds in the outermost domain for the 1:3 ratio experiments (50 and 750 seconds for the 1:5 ratio). For the smaller domains, the time steps are reduced by the factor of the nesting ratio.

Different physical parameterization schemes have been tested, in order to improve the representation of precipitation over Kenya. Tests have been done, by varying the cumulus, the long-wave (LW) radiation and the planetary boundary layer (PBL) parameterization schemes. For cumulus parameterization the Kain-Fritsch (Kain, 2004) and the scale-aware Grell-Freitas en-

semble (Grell and Freitas, 2014) schemes have been used, and one experiment is performed without using any cumulus parameterization at all. The LW radiation scheme has been varied between Rapid Radiation Transfer Model (RRTM; Mlawer et al., 1997) and Community Atmosphere Model (CAM; Collins et al., 2004). The two first-order non-local closure PBL schemes of Yonsei University (Hong et al., 2006) and the second version of the Asymmetric Convection Model (Pleim, 2007) have been tested. Table 1 provides the exact details of each experiment and the used parameterization options. The rest of the parame-

terization options are kept constant throughout the different experiments, i.e., WRF Single-moment 6-class scheme (Hong and Lim, 2006) for microphysics, Dudhia short-wave (SW) scheme for the SW radiation (Dudhia, 1988) and the Noah–MP land surface model (Niu et al., 2011; Yang et al., 2011) to describe surface processes. In all the simulations the lake model is turned on. The 1-D physically based lake model (Subin et al., 2012) helps to simulate lake internal processes and interactions at the surface of the lake with the atmosphere (Gu et al., 2015). It increases the eddy diffusivity and thus, it strengthens also the heat

transfer in the lake column (Gu et al., 2015). This is considered to be beneficial for the description of the lake surface temperature, which again helps to better represent evaporative effects and thus precipitation over the lake and in the surrounding areas. Please note that the aforementioned parameterization options are selected from an even larger set of experiments not included in this paper, tested with one nesting ratio and with four nested domains only.

As already shown in Table 1, each experiment obtained a name, chosen based on the area used in the literature or on the

main parameterizations that it employs. The "Europe" experiment is based on the parameterization options used with WRF over Europe in previous studies by the authors (Messmer et al., 2017), but including some updated schemes such as the Noah-MP land surface model. "South America" is based on the parameterizations used for the optimal simulation of storms over the central Andes (Zamuriano et al., 2019). The remaining "Cumulus" experiments are similar to the configurations applied over East Africa in previous studies (Pohl et al., 2011; Otieno et al., 2019), but they include the updated Noah-MP land

surface model, and changes in the cumulus scheme option (option 3 in WRF - "Cumulus3" experiment) or no use of cumulus parameterizations at all ("No Cumulus").



## 2.2 ERA5

ERA5 is the latest reanalysis provided by the European Centre for Medium-Range Weather Forecasts (ECMWF). At the moment, it is available from January 1979 until three months before present (Copernicus Climate Change Service (C3S), 2017). ERA5 provides different variables on the surface and various pressure levels with an hourly output. Nevertheless, we use 6-hourly data for our boundary conditions. The data are available globally on a 0.25° horizontal grid spacing and it is using 137 vertical model levels. A vast number of observations and satellite data are assimilated to the ERA5 gridded data using the integrated forecasting system cycle 41r2.

For the initial conditions, the surface variables surface and mean sea level pressure, 10 metre U and V wind components, 2-metre, dew point, skin and sea surface temperature (SST), and the volumetric soil water and soil temperature in four different levels are used. Additionally, some three-dimensional variables are used for initial and boundary conditions, such as the relative humidity, the U and V components of wind, the temperature and the geopotential. Only 24 out of the 137 vertical levels available in ERA5 were fed to WRF (1000, 925, 900, 850, 800, 775, 750, 700, 650, 600, 550, 500, 450, 400, 350, 300, 250, 200, 150, 100, 50, 30, 20, 10 hPa).

As stated before, our analysis will focus only on the year 2008. Compared to the climatology of Kenya for the year 1981–2010, the selected year is one of the warmer years, but when considering the constant warming since the beginning of the current millennium, it can be considered as a new normal (Fig. 2a). In terms of precipitation, 2008 is on the dry side compared to the climatology (Fig. 2b), but it is a year with two clear rainy seasons, the 'long rains' and the 'short rains' (Fig. 2c).

## 2.3 Observational data sets

To analyse the output of the WRF simulations and to identify the best parameterization options, the downscaled product must be compared to some independent observational data sets. Hence, the precipitation results are compared to ERA5, three satellite based data sets and individual weather station measurements. For temperature, the results are not only compared to ERA5, but also to the CRU data. Please note that the gridded data sets are interpolated to the grid of the WRF domain that it is compared to, and the nearest point to the station is considered afterwards. Consequently, small differences can appear in the values related to the gridded observational data sets. In the following, the different products are described in more detail.

### 2.3.1 Tropical Rainfall Measurement Mission (TRMM)

The Tropical Rainfall Measurement Mission (TRMM) comprises several data sets based on satellite data, and it is provided by NASA and the Japanese Aerospace Exploration Agency (JAXA). In this study we use the gridded data product TRMM 3B42 for precipitation estimates. Note that we use here the research-grade TRMM 3B42 and not the near real time version, as the first is considered to be more suitable for research (Liu, 2015). Version 7 of TRMM 3B42 is a combined product and merges satellite rainfall estimates with gauge data. To obtain the 3-hourly precipitation estimates, radars are calibrated to the microwave imager precipitation, which should result in a 3-hour microwave-only best estimate. In a next step, infrared precipitation is calibrated to the microwave product to fill regional gaps. Finally, the 3-hourly estimate is summed up to monthly values and re-calibrated





using a rain gauge analysis (Huffman et al., 2007, 2010). This monthly surface precipitation gauge analysis is obtained from

the Global Precipitation Climatology Centre (GPCC). The result is a Level 3 product with 3-hourly temporal and $0.25° \times 0.25°$ spatial resolution on a quasi-global ($50°$ N – $50°$ S) grid. With this resolution the TRMM 3B42 data is very similar to ERA5. TRMM 3B42 is available for the period 2000-02-29 to 2020-01-02, but for this study we only employ the year 2008.

### 2.3.2 IMERG

The Integrated Multi-satellitE Retrievals from GPM (IMERG) provides a multi-satellite product, currently available in the

$6^{th}$ version. It is the successor of the TRMM data set. Several products with different latency periods are available, but for this study we make use of the final product, which is suitable for scientific purposes. Similar to TRMM, several microwave measurements are used to estimate precipitation, and it is further calibrated against instrument products. The half hourly precipitation estimates are further recalibrated with a CMORPH Kalman filter and the PERSIANN Cloud Classification System artificial neural network. The product is finally adjusted to the monthly GPCC rain gauge measurements and is available in

half-hourly time steps and on a spatial resolution of $0.1° \times 0.1°$ (approximately 10 km $\times$ 10 km). The available time period is from June 2000 until present, but with a time lag of approximately 3.5 months. Here, we again only use the year 2008.

### 2.3.3 CHIRPS

The Climate Hazards group Infrared Precipitation with Stations (CHIRPS V2.0) provides a high-resolution data set with daily rainfall amounts (Funk et al., 2015). The $0.05°$ spatial resolved data are available for parts of the mid-latitudes and the trop-

ics ($50°$ S – $50°$ N). The data set is generated using thermal infrared precipitation products from different institutions. To calibrate global cold cloud duration rainfall estimates, the Tropical Rainfall Measuring Mission Multi-satellite Precipitation Analysis version 7 (TMPA 3B42 v7) is used (Funk et al., 2015). In a first step, the World Meteorological Organization's Global Telecommunications System (GTS) rain gauge data, which are relatively sparsely available, are combined with cold cloud duration derived precipitation estimates. In a second step, the best available weather station data are combined with cold cloud

duration based precipitation to get a product that on a monthly mean is similar to those produced by the GPCC or the Climate Research Unit from the University of East Anglia (Funk et al., 2015). Note that also IMERG and TRMM recalibrate their monthly output to results obtained from GPCC, and hence, the three data sets used for the model verification are not fully independent of each other, especially not when monthly sums are investigated.

### 2.3.4 CRU

The Climatic Research Unit (CRU) gridded time series of mean 2-metre temperature provides monthly data over land, except for Antarctica (version 4.03; Harris et al., 2020). The temperature is gridded on a $0.5 \times 0.5°$ resolution map and is based on a large number of weather stations. The weather station data are anomalized using their 1961-1990 climatology. These anomalies are gridded using an angular-distance weighting, and then they are converted back to absolute values. CRU gridded time series is often used in studies on African climate (e.g., Ongoma and Chen, 2017; Ayugi et al., 2020).





### 2.3.5 Weather station data

Compared to other tropical areas in Africa, Kenya and especially the area around Mount Kenya is covered by a comparably large number of weather station data with long precipitation measurement series. Many of these measurement series are maintained by farmers. Thanks to the support and involvement of the University of Nairobi and Bern, these series are still available today (Gichuki et al., 1998; Liniger et al., 2005; MacMillan and Liniger, 2005). In addition to private stations, there are also some that are operated by the government of Kenya, i.e., the Kenya Forest Service or the Kenyan Meteorological Department. We use data of 28 stations for precipitation which have been quality controlled by Schmocker et al. (2016). Table 2 provides some information on the stations used in this study, but for more detailed information the reader is referred to Table 1 in Schmocker et al. (2016). The station ID in both tables are identical to ease comparison. We obtained four stations with temperature records from the Social Hydrological Information Platform (SHIP), associated with the Water and Land Resources Centre (WLRC) project of the Centre for Training and Integrated Research In ASAL Development (CETRAD). Three additional stations for precipitation and temperature are included in the World Weather Records (WWR) database from the World Meteorological Organization (WMO). These are the three first lines of Table 2.

## 3 Results

In the following section we are presenting results of our various sensitivity experiments. The focus is on precipitation, as this variable is rather complex and still difficult to properly represent by regional climate models. Additionally, we present results from the innermost domain, i.e., the 1 km domain, as this is the region of interest. To obtain a complete picture of the best parameterization setup and nesting option, also the evaluation of temperature is presented and described.

### 3.1 Sensitivity of Precipitation

To investigate the sensitivity of simulated precipitation due to different parameterization options of the WRF model, we first show the annual cycle based on monthly means. Thereby, the sensitivity simulations with WRF and the three gridded observational data sets are compared to in-situ data from weather stations (see Table 2 for more details). To compare gridded data with point measurements at weather stations, the grid point that is closest to the corresponding latitude and longitude of the weather station is considered in the WRF simulation and the gridded observations. Two performance measures for each weather station are calculated and summarized in box-whisker plots (Fig. 3): the temporal correlation and the root mean squared error (RMSE). Additionally the standard deviation of each data set is compared to the one extracted from the weather stations (not shown). Several different gridded observational data sets are employed here to compare the sensitivity simulations and to classify which WRF setting performs best. As not only the weather station data but also the gridded observational data sets are subject to a range of uncertainties, we do not only rely on one product. Note that because the gridded data sets are interpolated to the respective WRF grid small differences can appear in the values of temporal correlations and RMSEs of each set-up and hence, also the shape of bars corresponding to these data sets in the box-whisker plots can look slightly different.





The temporal correlations show that the observational data sets, ERA5, TRMM and especially IMERG, are well correlated (Fig. 3a). This is expected as the data are not fully independent from each other. IMERG has the best correlation with the highest median but also with the smallest spread. This is true for all the different nesting options, but especially the WRF setting with a parent grid of 27 km and 4 domains shows a correlation of around 0.8 in the median value for all the observational data sets

. The fact that IMERG and the weather station data show such a good agreement further confirms the quality of the latter. The temporal correlations of the sensitivity simulations show a strong dependence on the nesting options. The simulations with fewer nests (right part of Fig. 3a) exhibit a higher correlation and a smaller spread than the two simulations that have one additional nest (left part of Fig. 3a). In particular, the "No Cumulus" simulation, but also the "Europe" parameterization, show a poor performance in the temporal correlation. Note that the poor performance of the "No Cumulus" can only be observed in

nesting options with a larger number of domains, i.e., setups with a parent grid of 27 or 25 km. The fact that the nesting option is important here suggests that with fewer domains (only 3 instead of 4 nests for the 1:3 ratio, and 2 instead of 3 for the 1:5 ratio), the simulation in the innermost domain is still more strongly influenced by the boundary conditions of the driving data, i.e., ERA5. Thus, the simulations with fewer nests cannot evolve with the same freedom as the ones with more nests, resulting in a better temporal agreement of the simulations. This is especially clear in the "No Cumulus" simulation.

While all the gridded observational data sets yield a rather high temporal correlation, the RMSE of ERA5 is rather high compared to the ones of TRMM, IMERG and CHIRPS (Fig. 3b). A reason for this is that the precipitation in ERA5 is independent of the weather station data, as precipitation is not assimilated into this product. Otherwise the RMSE shows similar results as the correlations for both the gridded observations as well as the sensitivity simulations. Hence, the parameterization of the simulation is only of minor importance compared to the nesting options. Similar findings are obtained when using the

standard deviation (not shown). Here, the WRF simulations are generally within the range of the standard deviation observed in the weather station data, except for the "Europe" parameterization in the nesting options with fewer nests. In that case, the standard deviation is strongly underestimated indicating that not the full variability of precipitation is captured. Additionally, the standard deviations of the gridded observational data sets are smaller than the ones of the weather station data, which is owed to the coarser resolution of the first.

Since the temporal correlation and the RMSE do not clearly define which parameterization option of WRF delivers the best results for precipitation in the region around Mount Kenya, we investigate the pattern correlation of the simulations compared to weather station data in a first step and to the gridded observational data set CHIRPS in a second step. Figure 4 shows in the first row the pattern correlation between the WRF-simulations and the weather station data for each month. The different columns indicate different parameterization options and the symbols within each panel shows the nesting option. The black

vertical line in each panel is equal to a correlation coefficient of 0.5. This value is a moderate correlation and still explains roughly 25 % of the variance, but it is a visual support to determine more easily which simulations and nesting options perform better than others. The number of months that are equal or exceed this limit of 0.5 in correlation are counted and summed up in the table below each panel ('# months' column).

The gridded observational data sets agree quite well in terms of the spatial pattern of precipitation, except for ERA5. The

fact that ERA5 shows a poor correlation with the weather station data is because the domain is located over steep terrain,

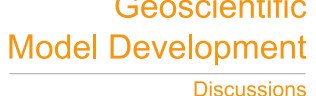

where a high resolution is needed to resolve precipitation patterns appropriately. Since ERA5 has the coarsest resolution of the here used observational data sets, it is not surprising that its correlations are worst. CHIRPS has the highest spatial resolution and shows a slightly better pattern correlation than IMERG (especially also in June) and hence, we have decided to compare the WRF-simulations also against the CHIRPS gridded data set (second row of Fig. 4). Similarly to the temporal correlation,

the simulations with fewer nests obtain a better pattern correlation compared to the ones with an additional nest. The "South America" and the "No Cumulus" parameterizations show the highest agreement with the weather station data for the nesting options that have an additional nest, while clearly the "Cumulus3 1-way" option is the best one of the simulations with fewer nests. Overall, the simulations have a better performance in the rainy seasons MAM and ON, while the dry months (and in particular June) are not very well captured by the model simulations.

Besides the comparison to the weather station data, the simulations and gridded observational data are compared to CHIRPS (see second row of Fig. 4). The gridded observational data sets perform well compared to CHIRPS (including also ERA5). Again this is expected as the data are not fully independent. The pattern correlation of the WRF simulations compared to CHIRPS are rather high in all the simulations. No clear difference between the different nesting options are evident. The "South America" and the "No Cumulus" options show the best agreement with CHIRPS in precipitation patterns, but also "Cumulus3

1-way" performs well. The "Europe" parameterization is clearly the worst, even if it shows one of the highest correlations in the dry months June and July. This is because the "Europe" parameterization setup produces rather dry conditions over Africa and hence, the dry months are better represented compared to the others that simulate generally wetter conditions.

To further understand how well the different parameterization and nesting options are able to represent precipitation around Mount Kenya, the annual cycle is plotted as grid point averages of monthly precipitation sums of the innermost domain (1 km;

Fig. 5). Please note that the gridded observational data sets do not obtain exactly the same values for the different nesting ratios 1:3 (first row) and 1:5 (second row), as the domain sizes are not exactly equal. The innermost domain in the 1:5 nesting ratio setup is slightly bigger. The three gridded observational data sets TRMM, IMERG and CHIRPS agree well and are considered as the reference here, as they show a good temporal and pattern correlation with the weather station data. This is true except for November, when CHIRPS records a much higher value in precipitation amounts than TRMM and IMERG. As CHIRPS

shows one of the weakest pattern correlations in November compared to the weather station data, IMERG and TRMM should be considered as the reference in this month. ERA5 does also agree, but the 'long rains' (MAM) have the peak intensity a bit too early, while the intensity in the 'short rains' (ON) are too intense on average. For the dry months, ERA5 also overestimates precipitation compared to the other gridded observational products. Overall the gridded data sets come up with similar annual precipitation sums (inset of Fig. 5), except for ERA5, which shows a slight overestimation in annual precipitation sums.

Comparing the monthly precipitation sums of the sensitivity simulations with the gridded observational reference, we find again that the "Europe" parameterization option is not well suited for this area as it is not able to capture correctly the two rainy seasons near Mount Kenya. The 'long rains' show a clear deficit in precipitation, while the outcome of the 'short rains' strongly depends on the number of nests. With fewer nests the 'short rains' are also clearly underestimated, but with an additional nest precipitation is either almost correct or overestimated. The differences between the "Europe" parameterization and the

others are the long wave radiation and the PBL parameterization, which are both responsible for the reduction in precipitation





amounts (not shown). The "No Cumulus" setup performs well in both wet seasons, but it overestimates precipitation during the dry season. The "Cumulus3" options show a clear sensitivity of the precipitation amounts in the 'long rains' to the number of nests, with a much better representation with fewer nests. In the 'short rains', the "Cumulus3" options follow the curve of CHIRPS and therefore shows an overestimation. The fact that the "Europe" setting is not suited for this regions becomes even

clearer by including the annual precipitation sums. Except for the 25 km parent grid, the "Europe" setting captures only around 50 % of the annual precipitation, and hence, it clearly underestimates the water availability. All the other settings perform similarly well on an annual basis. It is further noteworthy that the WRF model with the "Cumulus 3" options is able to correct the overestimation obtained by ERA5, which is the driving data set of the simulations.

Another measure used to identify the best setup of WRF for this region are the precipitation patterns of the WRF simulations.

While the parameterizations are tested using ERA5 as boundary condition in order to be able to compare it against observations of year 2008, the final goal is to apply the WRF setup to climate simulations, which normally have a coarser resolution (around 100 km) than reanalysis data (around 30 km for ERA5). In that case, a climate simulation with a parent domain starting at 9 or 5 km is not possible. Therefore, in the following, the simulation of the innermost domain (D4; 1 km) with the parent domain of 27 km horizontal grid spacing and a nesting ratio of 1:3 is presented. Note that the simulations with a 25 km parent grid

and a nesting ratio of 1:5 show similar results and hence, are not shown here. As CHIRPS is the data set that shows the best agreement with the weather station data (as shown in Fig. 4), and it also shows the highest resolution and detail, the patterns will be mainly compared to this data set. To better understand how resolution can affect the representation of precipitation we also show IMERG and ERA5. The latter also helps to see the added value of regional climate models. Three months are picked to present the results: April (Fig. 6) and November (Fig. 7) as they are within the 'long rains' and 'short rains', respectively

and June (Fig. 8) because it represents a rather dry month, which is difficult to capture by the WRF simulations.

In April, CHIRPS is able to simulate correctly the measured amounts of precipitation, with the exception of a small region located to the north of Mount Kenya where precipitation is overestimated by this gridded dataset. IMERG, but also ERA5 are too coarsely resolved to capture the fine scale structure in precipitation patterns in this complex topographic area. While IMERG presents a similar structure to CHIRPS, ERA5 is just highlighting Mount Kenya. Baring in mind that also the observational

data sets are subject to some uncertainties, it is still obvious that the "Europe" parameterization is too dry (as previously noted) compared to all the gridded observational data sets in April. It also shows a diagonal band southeast of Mount Kenya, while CHIRPS indicates that the whole lower right corner receives some similar amount of precipitation. This diagonal band is also present in the "South America" parameterization. The precipitation amounts are in some areas higher compared to CHIRPS, but generally more stations agree with the pattern obtained with this parameterization option than the "Europe"

setting. The other three options manage to produce a precipitation pattern as observed in CHIRPS. Nevertheless, the "No Cumulus" parameterization is too wet especially in the northwestern part of the domain, and the steep gradient from high precipitation rates in the vicinity of Mount Kenya to dryer conditions to the northwest of it is not well captured. The two "Cumulus3" parameterization options capture this pattern the best, including also some finer details along the right and bottom boundaries of the domain.





CHIRPS is capturing the precipitation pattern quite well in November, with some deviations compared to the weather station data south and northwest of Mount Kenya (Fig. 7). IMERG provides a rather homogeneous precipitation pattern, lacking much of the finer scale structure. ERA5 obtains a similar precipitation pattern as in April, featuring intense precipitation around Mount Kenya. This leads to a wrong representation in precipitation amounts along the steep northwestern slopes of Mount Kenya. In this month, the "Europe" setting is again a bit too dry, capturing only the precipitation patterns around mount Kenya, and not showing a good performance in the surrounding flatlands. " South America" captures the precipitation amounts of the stations quite well, but the pattern shows some deviations compared to CHIRPS, especially in the northwestern corner of the domain. The two "Cumulus3" and the "No Cumulus" options are able to capture the patterns well with some overestimation in the simulations driven by the first options and a slight underestimation in precipitation amounts for the latter parameterization option. Given the uncertainty range within the observation based data, it cannot be expected that a single sensitivity simulation can agree with all the stations or with one of the gridded observational data sets.

June is clearly much drier than April and also CHIRPS receives too much of precipitation compared to the weather station data (see Fig. 8). Nevertheless, it again captures the pattern of the stations best, as IMERG misses peak precipitation south of Mount Kenya and ERA5 produces overall too high precipitation amounts. It is obvious that ERA5 generates in all the months a very similar precipitation pattern, where only the amount of precipitation is changing. Given the uncertainty range of the observation based data, the "Europe" parameterization option is too dry in some areas and too wet in others. The "South America" parameterization option obtains at most of the stations too much precipitation, while also the general pattern is a bit off, as no dry corridor in the east can be observed. This is also true for the "No Cumulus" parameterization option, but here the pattern agrees better, with a clear overestimation in precipitation amounts. "Cumulus3" and "Cumulus3 1-way" again result in the best pattern rendering it difficult to choose between the two, as some stations are better in one and other stations are better described in the other setting. Since one-way nesting does not overwrite the solution of the corresponding parent grid, this option should be preferred over the two-way nesting option. It allows to not only focus on the innermost domain, but also investigations of a larger scale picture are able without any disturbances within the domain.

All the WRF simulations reasonably resemble the precipitation pattern over Mount Kenya in the year 2008. The "No Cumulus" parameterization and the "Cumulus3" options provide throughout the analysis the best performances. The fact that the "No Cumulus" option is generally too wet allows us to define the "Cumulus3 1-way" option as the best for our purpose. Note also that the "No Cumulus" parameterization option produces a patchy picture in the outermost domain with monthly sums, which is a clear sign of a structural problem, i.e., convection is induced always at the same location (not shown), which is rather unrealistic. Hence, this simulation is also not suitable for a larger scale analysis of precipitation and precipitation changes in a warmer climate.

## 3.2 Sensitivity of Temperature

Once we have investigated the sensitivity of simulated precipitation due to different parameterization options, we focus on temperature. To do so, the sensitivity simulations with WRF are compared to the driving reanalysis ERA5, the gridded observational data set CRU and the in-situ data from weather stations (see Table 2 for more details). To measure how the different





settings simulate the temperature near Mount Kenya, the 2-metre temperature patterns are evaluated. The same months as for
precipitation are selected for temperature: April (Fig. 9) and November (Fig. 10), within the 'long' and 'short rains', respectively, and June (Fig. 11), within the dry season. In order to highlight the differences between the sensitivity experiments, only the absolute values of the gridded data sets and the "Cumulus3 1-way" experiment are depicted. For the remaining sensitivity experiments, the anomalies compared to our best setting experiment are presented. Note that the weather station data have not been adjusted to the height of the model topography as the differences in height are in the range of a few meters. The maximum
difference between the station and model height is around 60 meters and hence the maximum discrepancy between station and modelled temperature is around 0.4 °C, if we consider the standard environmental lapse rate of 6.5 K/km (Barry, 2008).

ERA5 temperature serves as boundary and initial conditions for the WRF simulation. Given this constraint, we expect a better representation of the simulated 2-metre temperature than of precipitation. Also, the region of interest is dominated by steep topography, which is directly related to temperature and consequently, strong gradients of temperature are expected near
Mount Kenya.

It is not surprising that ERA5 and also CRU represent temperature relatively similarly and independently of the season (rainy or dry) as Kenya is located at the equator. ERA5 describes the orography of the domain a bit more clearly, because the resolution is finer. Most of the weather station data agree well with ERA5 and CRU, but the WRF simulation "Cumulus3 1-way" produces a better picture of the temperature profile, which is mainly owed to the better resolution and a more detailed
characterization of the topography.

In April (Fig. 9), the two "Cumulus3" simulations have a very similar representation of temperature, as the only difference is the communication between the nests. The differences for the "South America" and "Europe" experiments are mainly in the range of ±0.5 °C, with positive anomalies over Mount Kenya and with negative anomalies in the surrounding plains. In the case of the "No Cumulus" parameterization, strong negative anomalies are observed over the entire region, but particularly in
the southeastern corner of it where anomalies of 2 °C are noticed. This observation of clearly cooler temperature is related to the overestimation in precipitation. Hence, most of the domain obtains more precipitation than in the "Cumulus3 1-way" simulation. This excess in water can be transformed into latent heating through evaporation and can contribute to a cooling effect over the domain.

In November the "Cumulus3 1-way" option overestimates the precipitation somewhat, which generally results in cooler
temperatures compared to observations in some stations (e.g., station 1 and 4 in Fig. 10). As the other sensitivity experiments simulate a drier monthly climate especially in the plains, a positive temperature signal can be observed in these areas as well. Generally, the temperature difference between the experiments is again rather small and below 1 °C.

The two "Cumulus3" options are able to simulate correctly the observed temperature in June (Fig. 11), and their differences are rather small (below ±0.25 °C). The biases for the "South America" and "Europe" experiments are more intense than in
April, but the patterns are similar (positive anomalies over Mount Kenya, but negative ones in the plains). Again, negative anomalies are obtained for the "No Cumulus" parameterization. This is also related to an increase in precipitation amounts, especially in the southeastern corner of the domain.





As described above, the differences in the patterns in the three months reveal more or less the differences in precipitation, i.e., where more precipitation is simulated compared to "Cumulus3 1-way", the temperature is cooler as more energy is trans-

formed into latent heating through evaporative processes. Where precipitation is comparably reduced to the "Cumulus3 1-way" parameterization, a warming can be observed as energy is transformed into sensible heating. Additionally, moisture advection but also small differences in the description of cloud cover can lead to some of the changes in temperature. The "Europe" simulation is the one experiment that shows the biggest discrepancy between the moisture availability and temperature response, which might be related to the fact that also the LW and PBL parameterizations schemes are different compared to the

"Cumulus3 1-way" option.

## 4   Summary and Conclusions

The goal of this study was to find a setup for WRF in order to realistically simulate precipitation patterns and amounts over and around Mount Kenya at a kilometer scale. This task is challenged by the fact that this region has a complex topographic structure and is influenced by large-scale circulation controls, which lead to heterogeneous precipitation patterns. As this is one

of the first studies to resolve Mount Kenya region at such fine scale, different parameterization options and combinations must be tested to obtain an optimal result for this area. We employ the WRF model and experiment with cumulus (Kain-Fritsch, Grell-Freitas, no cumulus), LW (CAM, RRTM) and PBL (ACM2, YSU) parameterizations, but also with the number of nested domains and nesting ratios (1:3 and 1:5). The different simulations are not only compared to different gridded observational data sets, such as IMERG, TRMM and CHIRPS, but also to a large number of weather station data operated by private farms,

CETRAD and the Kenyan government.

Correlating the annual cycle as monthly sums reveals that the gridded observations and the weather station data agree very well, indicating that the here presented weather station data are reliable. The temporal correlations further lead to the conclusion that if ERA5 is used as boundary conditions in a smaller and higher resolved domain (i.e., simulations with one less nest) the simulation is more constrained and hence, the temporal correlation is better with a reduced number of nests.

This result is mainly important for simulations driven with reanalysis data, as they capture most of the atmospheric circulation and processes well and therefore are reliable, which is not necessarily true for climate simulations as well. The "No cumulus" parameterization scheme is especially sensitive to changes in number of nests in terms of temporal correlation. Concerning the nesting ratio, we were not able to distinguish between the two options, so any of the two are able to produce realistic results.

Also important for water availability in the area around Mount Kenya are the precipitation patterns and amounts. The

objective pattern correlations indicate that also fewer nests result in a better spatial correlation, but when comparing against the most accurate gridded data set, CHIRPS, there is not much difference between the number of nests in spatial patterns. Compared to the temporal correlation, the "No Cumulus" parameterization results in rather accurate pattern correlations. The pattern correlation is not only a valuable tool to evaluate the sensitivity simulations, but also the gridded observational data sets. The comparison to the weather station data reveals that CHIRPS yields a pattern closest to the weather stations. One important



factor for this result is certainly the nominal resolution of the data, as CHIRPS reveals the finest precipitation structure of all the gridded observational data sets.

The "Europe" configuration obtains not only one of the worst temporal, but also pattern correlation. The actual patterns within the innermost domain reveal that the "Europe" configuration is clearly too dry, both in the rainy and the dry seasons. The underestimation in precipitation can be attributed to both the LW and PBL parameterizations. But not only the precipitation

amounts are underestimated, also the pattern is not fully captured. The "South America" setting is more accurate when it comes to monthly precipitation sums in the rainy season, but it clearly has a wet bias in the dry season, and also the pattern is missing some details compared to CHIRPS. While the two "Cumulus3" options and the "No Cumulus" option provide rather good patterns, the latter clearly overestimates the monthly sums. Hence, we conclude that the "Cumulus3 1-way" option is the best parameterization setting in WRFV3.8.1 for the area around Mount Kenya. The 1-way option is preferred over the 2-way

option, as the latter affects the representation of the domain when cumulus parameterization is turned off. Hence, with the 1-way option, all domains and scales of the simulation can be integrated into the analysis.

Similar to other studies, we also find an overestimation in precipitation compared to observations (Cook and Vizy, 2013; Williams et al., 2015). Nevertheless, with our sensitivity studies we were able to find a parameterization option that represents precipitation amounts rather well in the rainy seasons, while a wet bias remains in the dry season. Certainly, the very high

resolution of our simulations helps to better represent not only the pattern of precipitation, but also of temperature, as also mentioned in Kerandi et al. (2017). It is not surprising that a high resolution can add value in the representation of precipitation and temperature, as this region is located within complex topographic structures.

Having found the optimal setting for the Mount Kenya area, climate change simulations can be performed. These allow to get a detailed picture of the climate sensitivity in this area and the possible changes in water availability and the actual warming

in the area. Furthermore, sensitivity experiments to land use changes can be created. This will help to understand how future changes in agriculture will affect water availability in the flatlands around Mount Kenya.

*Data availability.* All the observational gridded data sets for precipitation included in this study are freely available online. TRMM and IMERG can be downloaded from the Earth Observing System Data and Information System (EOSDIS) from NASA (https://doi.org/10. 5067/TRMM/TMPA/3H-E/7 and https://doi.org/10.5067/GPM/IMERG/3B-HH/06 respectively), and CHIRPS can be download from the

Climate Hazards Center of the UC Santa Barbara (https://doi.org/10.15780/G2RP4Q). CRU data set used in the analysis of temperature is also available online and can be downloaded from the Climatic Research Unit of the University of East Anglia (https://crudata.uea.ac. uk/cru/data/temperature/). The weather station data from WMO used in this study can be downloaded from the World Weather Records website (https://www.wmo.int/pages/prog/wcp/wcdmp/GCDS_2.php). The data from the stations maintained by CETRAD in Kenya can be downloaded from the Social Hydrological Information Platform (http://www.wlrc-ken.org/admin/dashboard/home). The postprocessed

outputs for precipitation and temperature from our WRF sensitivity experiments can be downloaded from: https://doi.org/10.5281/zenodo. 4090589.





*Author contributions.* The conceptualization was developed by all the authors. The preparation of data sets and the methodology was designed by M.M. and S.J.G.-R. The analysis was carried out by all the authors. The original draft of the paper was written by M.M. and S.J.G.-R., but all the authors took part in the edition and revision of it.

*Competing interests.* The authors declare no competing interests.

*Acknowledgements.* The authors thank the Wyss Foundation for funding this pilot project. The authors acknowledge CETRAD, Noemi Imfeld and Stefan Brönnimann for sharing the weather station data of Kenya. Financial support by the Swiss National Science Foundation (Early PostDoc.Mobility grant: P2BEP2-181837) and the Oeschger Centre for Climate Change Research is acknowledged. The TMPA data were provided by the NASA/Goddard Space Flight Center's Mesoscale Atmospheric Processes Laboratory and PPS, which develop and
compute the TMPA as a contribution to TRMM. The computational resources were provided by CSCS, and the authors thank the creators of the WRF model. Finally, some of the calculations were carried out with R (R Core Team, 2018), and the authors want to thank all the authors of the packages used for it: akima, ggplot2, latticeExtra, maptools, plotrix, reshape2, rgdal, rgeos, RNetCDF, shape and sp.





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



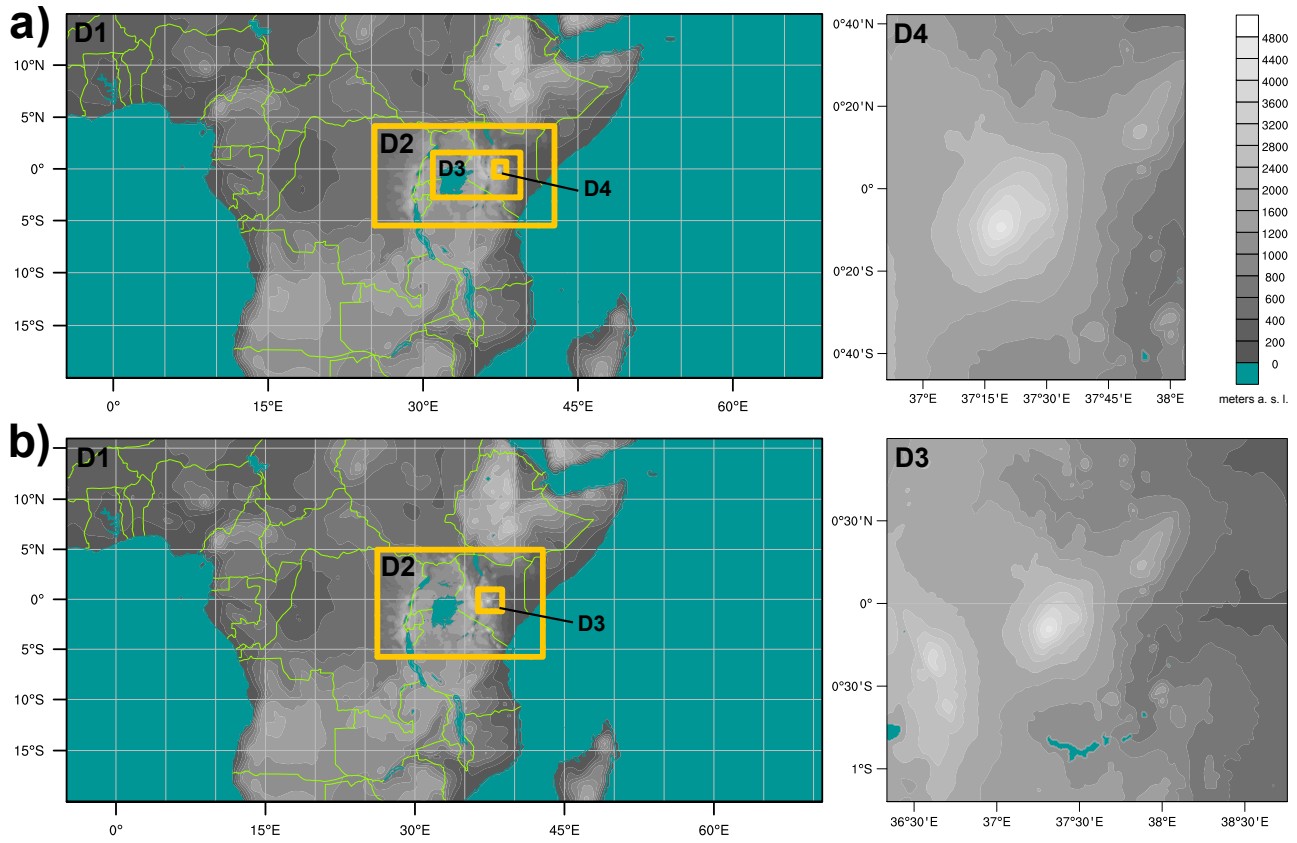

**Figure 1.** The two different nesting settings for the sensitivity experiments are depicted. The domains of the nesting ratio 1:3 are shown in row a) with the topography of the innermost nest D4 in the right panel. The domains of the nesting ratio 1:5 are shown in row b) with the topography of the innermost nest D3 in the right panel. D1 = 27/25 km, D2 = 9/5, D3 = 3/1, D4 = 1 km. The grey shading indicates elevation in meters above sea level using the WRF topography Global Multi-resolution Terrain Elevation Data (GMTED2010) provided by USGS.



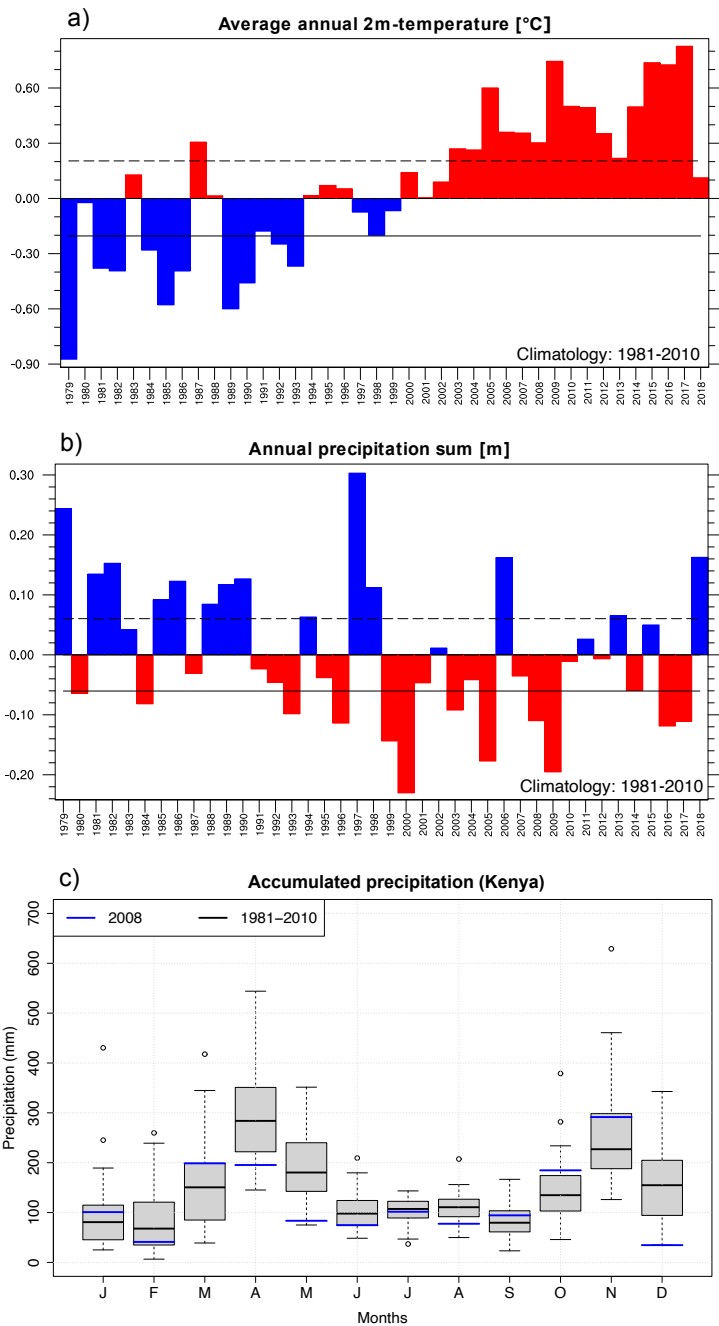

**Figure 2.** Annual mean 2m-temperature (a) and precipitation (b) anomalies from ERA5 (Copernicus Climate Change Service (C3S), 2017). The anomalies are calculated with respect to the climatological mean of the years 1981 to 2010. The stippled (straight) lines illustrate plus (minus) one standard deviation. Monthly accumulated values of precipitation (in mm) for the selected year 2008 (in blue), compared to the climatology (1981-2010, in grey, using a Box-Whiskers plot) are shown in (c). All values are means over the territory of Kenya in each subplot. The whiskers extend to the value that is no more than 1.5 times the inter-quartile range away from the box. The values outside this range are defined as outliers and are plotted with dots.

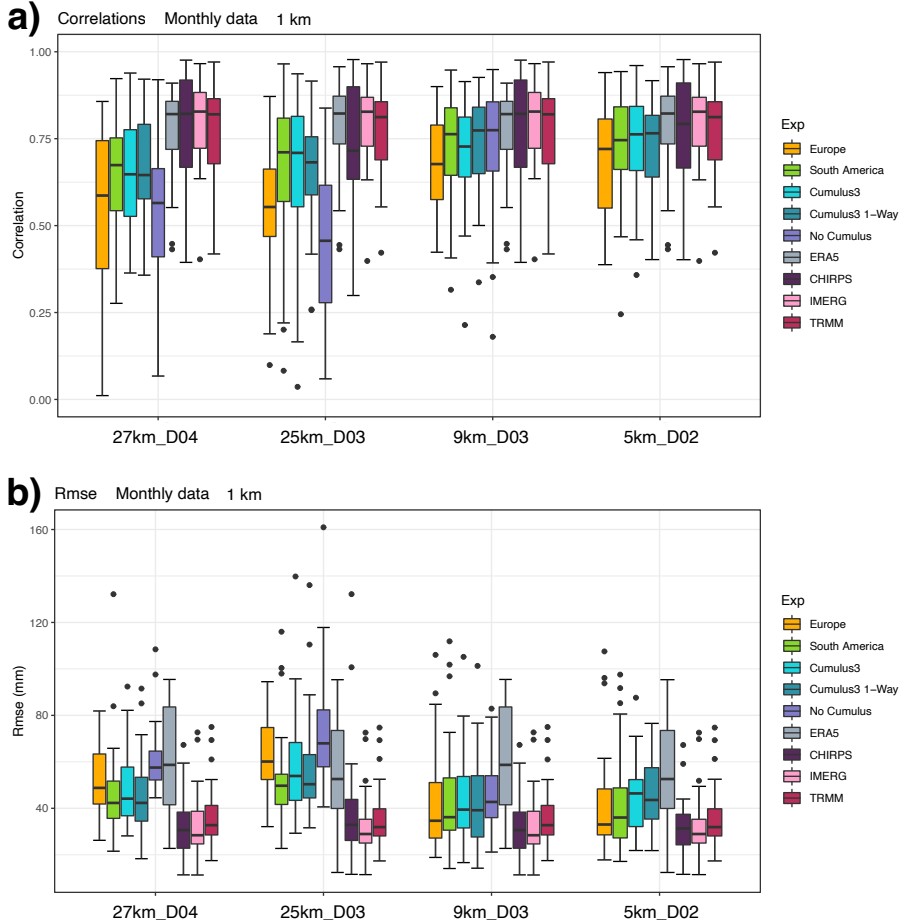

**Figure 3.** The temporal correlation (a) and root mean square error (RMSE) (b) between the annual cycle of measured and simulated monthly precipitation sums at the nearest grid point to the station's location are shown for the different parameterization options (see legend to the right, Table 1), grouped by the different nesting options and number of nests. The box and whisker plots show the values in relation to 28 stations for the different domains with 1 km spatial resolution. The whiskers extend to the value that is no more than 1.5 times the inter-quartile range away from the box. The values outside this range are defined as outliers and are plotted with dots.

**Figure 4.** Pattern correlation of monthly precipitation sums between weather station data and the respective WRF simulation (upper row), and between CHIRPS (interpolated onto the respective WRF domain) and the respective WRF simulation (lower row). The different panels indicate the different parameterization options (Table 1), and the symbols stand for the different nesting options. The labelling of the symbols is given in the table below each panel, along with the number of months (# months) in which the nesting option obtain correlation patterns above the reference value of 0.50 (a moderate correlation used to visually evaluate the performance of nesting options). The last panel on each row represents the gridded data sets used throughout the paper. Even if the gridded data sets are interpolated onto different domains for each independent set-up, here only one setting is shown (27 km_D4). The rest was omitted as only marginal changes can be observed.



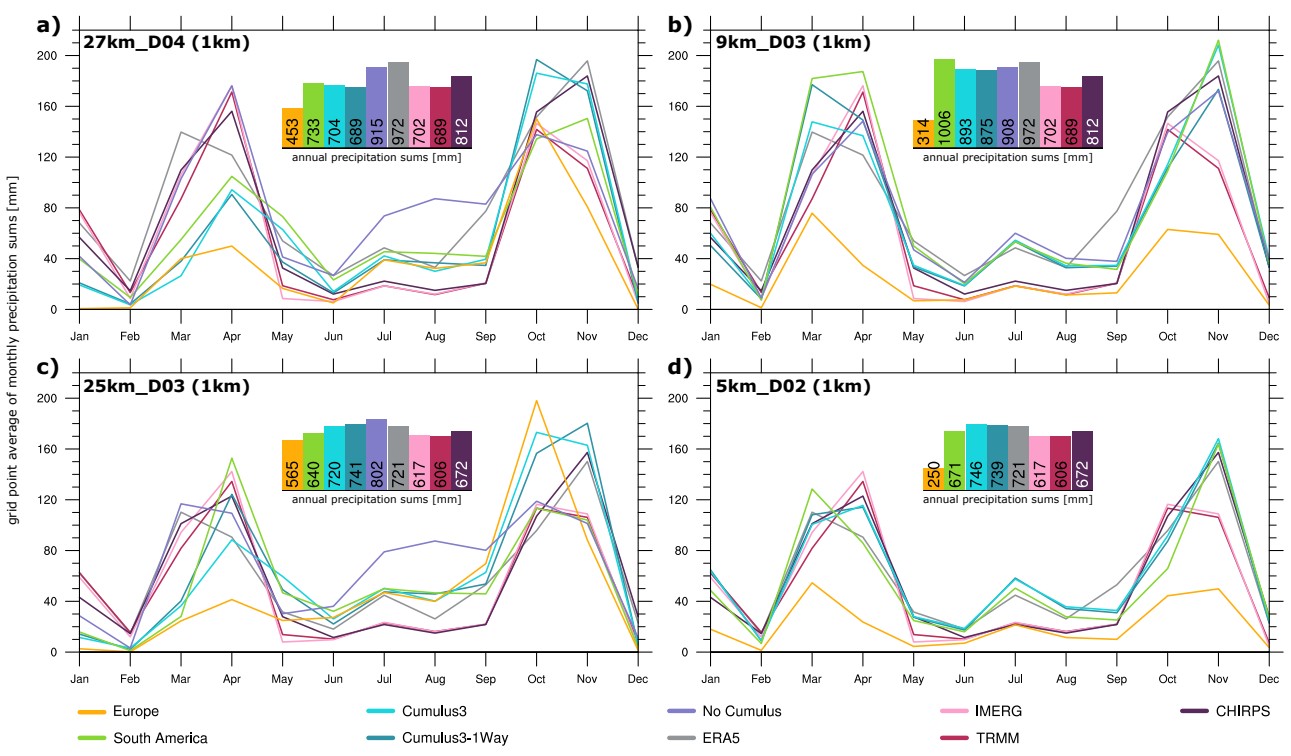

**Figure 5.** Grid point averages of monthly precipitation sums of the innermost domain (1 km) for each of the tested set-ups: 27 km (a), 9 km (b), 25 km (c) and 5 km (d). The 5 tested parameterization options are included, along with the driving reanalysis ERA5 and the three observational gridded data sets (IMERG, TRMM and CHIRPS). All the gridded data sets are plotted with different shades of pink, while ERA5 is colored in grey. The inset of a bar plot in each panel indicates the grid point average annual precipitation sum for each parameterization option and gridded data set.



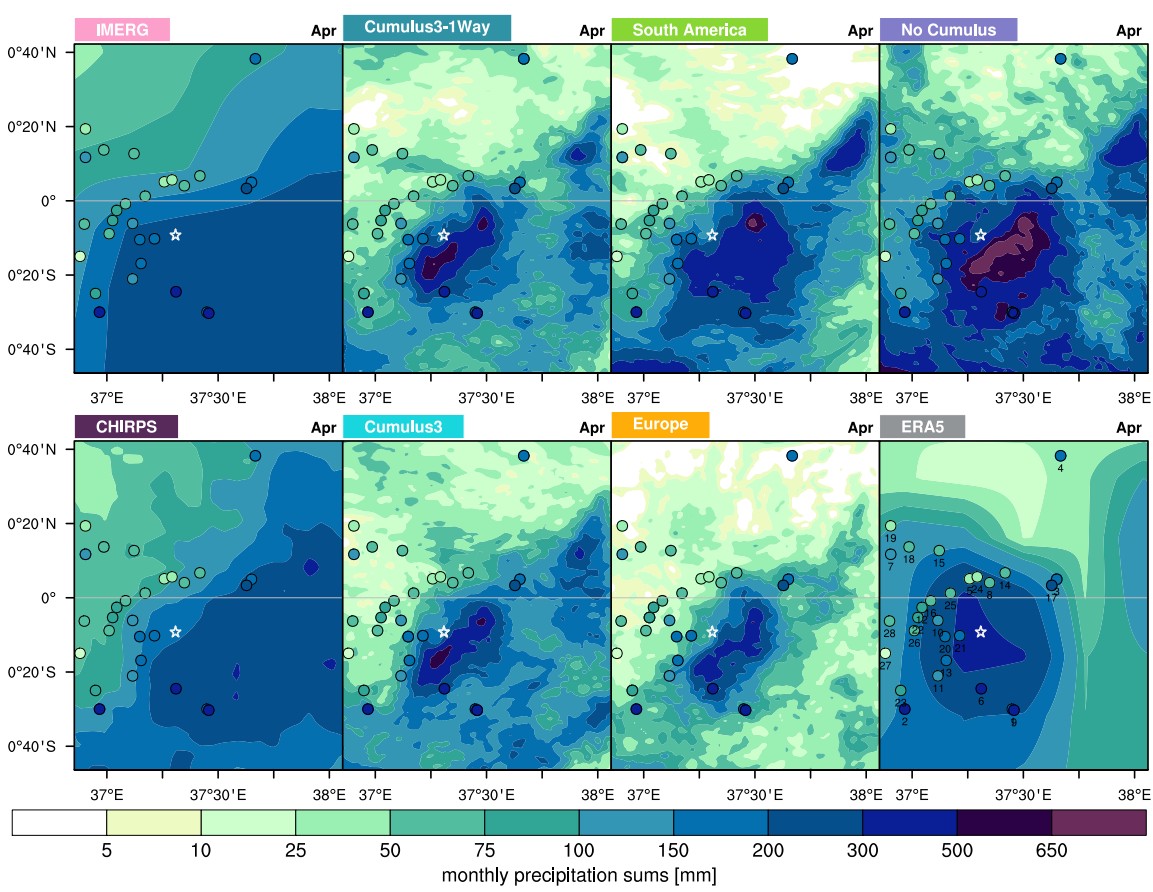

**Figure 6.** Monthly precipitation sums for April 2008 ('long rains') in mm for the innermost domain (1 km) of the 4 nested domain setup, with the outermost domain of 27 km resolution and a nesting ratio 1:3 for the different parameterization setups (see Table 1). Weather station data are described in Table 2. The white star indicates the summit of Mount Kenya.



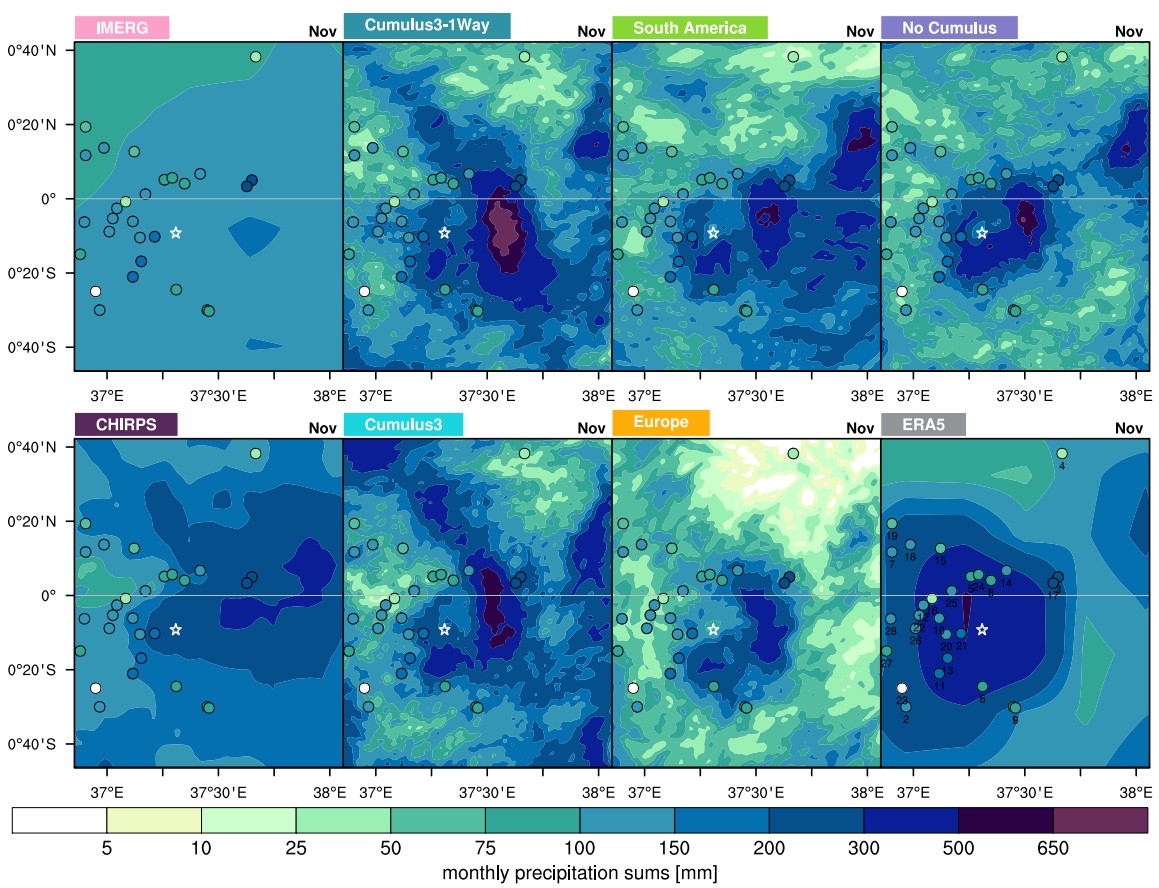

**Figure 7.** Same as Fig. 6 but for November 2008 ('short rains').





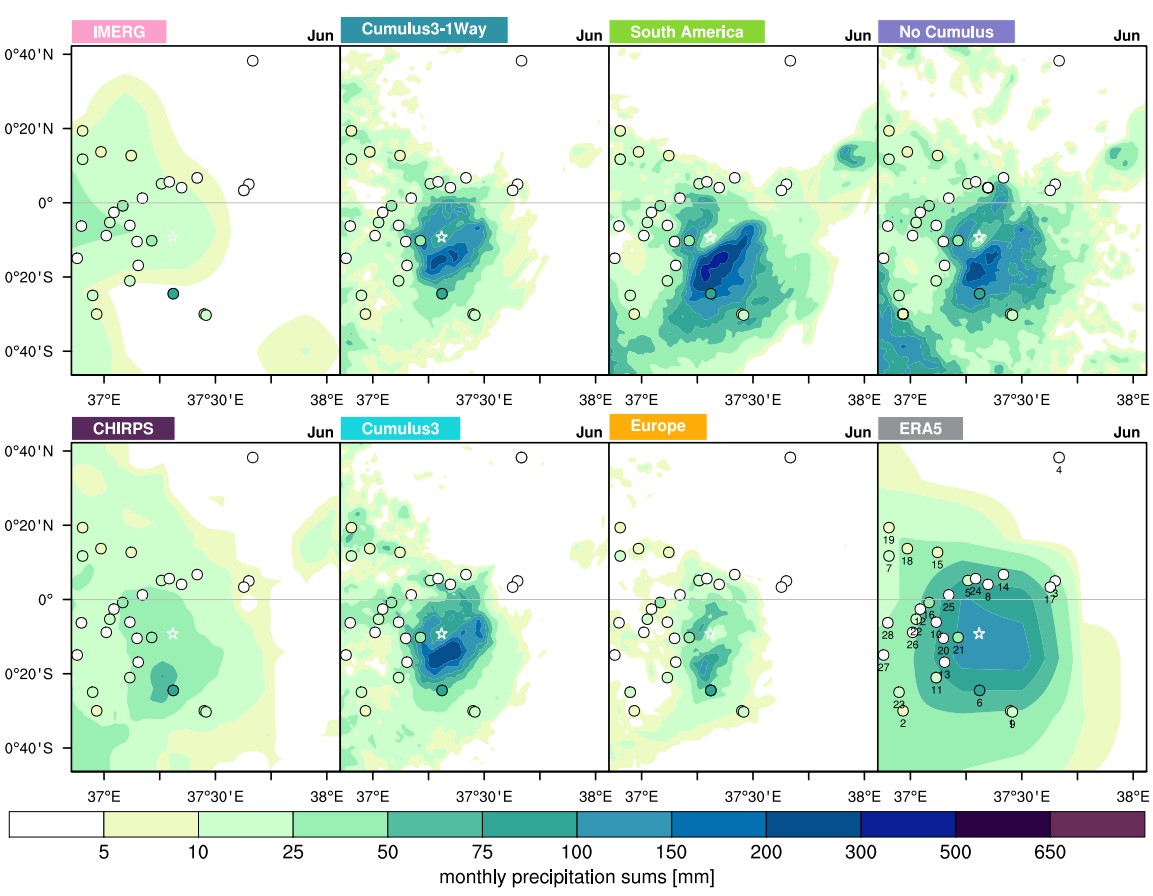

**Figure 8.** Same as Figs. 6 and 7 but for June 2008.





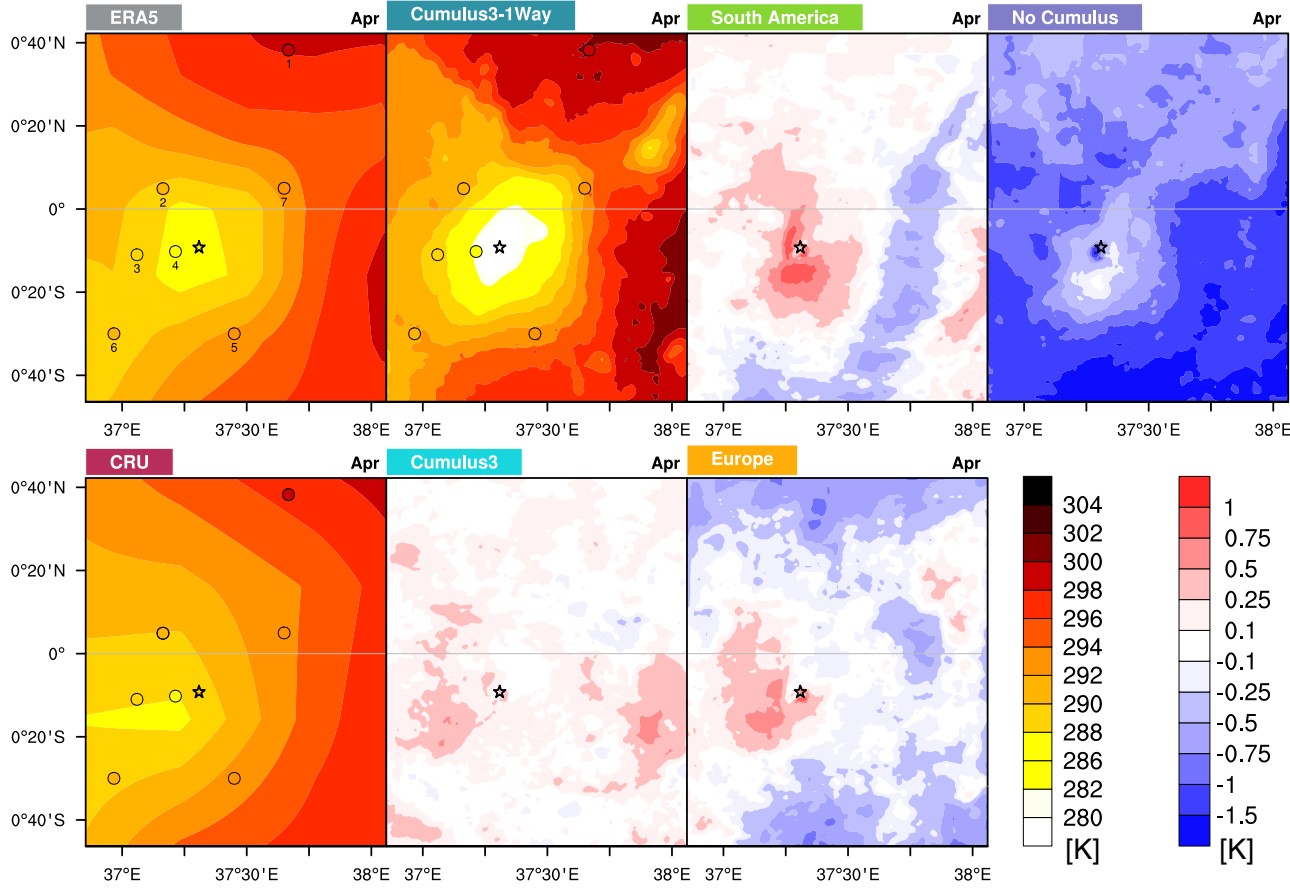

**Figure 9.** Monthly 2-metre temperature averages for April 2008 ('long rains') in K are shown for the innermost domain (1 km) of the 4 nested domain setup, with the outermost domain of 27 km resolution and a nesting ratio 1:3 for the different parameterization setups (Table 1). Absolute values are given for ERA5, CRU and the "Cumulus3 1-way" option. The others depict anomalies compared to the "Cumulus3 1-way" option. Weather station data are described in Table 2. The black star indicates the summit of Mount Kenya.





**Figure 10.** Same as Fig. 9 but for November 2008 ('short rains').





**Figure 11.** Same as Figs. 9 and 10 but for June 2008.





**Table 1.** Experimental design of the sensitivity simulations: Name of the experiment, parameterizations used and other parameters important for each experiment such as nesting option and ratio, number of domains, names of domains as they appear in Fig. 1 and their corresponding spatial resolution. The last column provides the name of the innermost domain, used in Figures to identify nesting and resolution options.

| Name | Parameterizations | | | Other parameters | | | | | Name |
| | Cumulus | LW-Rad. | PBL | Nesting | Nest ratio | # dom | doms in Fig. 1 | Resolutions | 1km domain |
|---|---|---|---|---|---|---|---|---|---|
| Europe | Grell-Freitas | CAM | ACM2[1] | 2-way | 1:3 | 4 | (a) D1, D2, D3, D4 | 27, 9, 3, 1 km | 27km_D04 |
| | Grell-Freitas | CAM | ACM2 | 2-way | 1:3 | 3 | (a) D2, D3, D4 | 9, 3, 1 km | 9km_D03 |
| | Grell-Freitas | CAM | ACM2 | 2-way | 1:5 | 3 | (b) D1, D2, D3 | 25, 5, 1 km | 25km_D03 |
| | Grell-Freitas | CAM | ACM2 | 2-way | 1:5 | 2 | (b) D2, D3 | 5, 1 km | 5km_D02 |
| South America | Kain-Fritsch | RRTM[2] | YSU[3] | 2-way | 1:3 | 4 | (a) D1, D2, D3, D4 | 27, 9, 3, 1 km | 27km_D04 |
| | Kain-Fritsch | RRTM | YSU | 2-way | 1:3 | 3 | (a) D2, D3, D4 | 9, 3, 1 km | 9km_D03 |
| | Kain-Fritsch | RRTM | YSU | 2-way | 1:5 | 3 | (b) D1, D2, D3 | 25, 5, 1 km | 25km_D03 |
| | Kain-Fritsch | RRTM | YSU | 2-way | 1:5 | 2 | (b) D2, D3 | 5, 1 km | 5km_D02 |
| Cumulus3 | Grell-Freitas | RRTM | YSU | 2-way | 1:3 | 4 | (a) D1, D2, D3, D4 | 27, 9, 3, 1 km | 27km_D04 |
| | Grell-Freitas | RRTM | YSU | 2-way | 1:3 | 3 | (a) D2, D3, D4 | 9, 3, 1 km | 9km_D03 |
| | Grell-Freitas | RRTM | YSU | 2-way | 1:5 | 3 | (b) D1, D2, D3 | 25, 5, 1 km | 25km_D03 |
| | Grell-Freitas | RRTM | YSU | 2-way | 1:5 | 2 | (b) D2, D3 | 5, 1 km | 5km_D02 |
| Cumulus3 1-Way | Grell-Freitas | RRTM | YSU | 1-way | 1:3 | 4 | (a) D1, D2, D3, D4 | 27, 9, 3, 1 km | 27km_D04 |
| | Grell-Freitas | RRTM | YSU | 1-way | 1:3 | 3 | (a) D2, D3, D4 | 9, 3, 1 km | 9km_D03 |
| | Grell-Freitas | RRTM | YSU | 1-way | 1:5 | 3 | (b) D1, D2, D3 | 25, 5, 1 km | 25km_D03 |
| | Grell-Freitas | RRTM | YSU | 1-way | 1:5 | 2 | (b) D2, D3 | 5, 1 km | 5km_D02 |
| No Cumulus | - | RRTM | YSU | 1-way | 1:3 | 4 | (a) D1, D2, D3, D4 | 27, 9, 3, 1 km | 27km_D04 |
| | - | RRTM | YSU | 1-way | 1:3 | 3 | (a) D2, D3, D4 | 9, 3, 1 km | 9km_D03 |
| | - | RRTM | YSU | 1-way | 1:5 | 3 | (b) D1, D2, D3 | 25, 5, 1 km | 25km_D03 |

[1] Asymmetric Convection Model Version2

[2] Rapid Radiative Transfer Model

[3] Yonsei University





**Table 2.** Weather station information: Station number used in our study (labels in Figs. 6-11), station name, location (latitude and longitude), altitude (in meters above sea level), number of missing values, variables available and station ID from WMO (first three lines) or from Table 1 in Schmocker et al. (2016). RR stands for precipitation and T2 for 2-metre temperature.

| Number | Station | Lat | Lon | Altitude [m] | # missing | variable | ID |
|--------|---------|-----|-----|--------------|-----------|----------|-----|
| 1 | Embu WMO | -0.5 | 37.45 | 1493 | 0 | RR, T2 | 637200 |
| 2 | Nyeri WMO | -0.5 | 36.967 | 1759 | 0 | RR, T2 | 637170 |
| 3 | Meru WMO | 0.083 | 37.65 | 1554 | 0 | RR, T2 | 636950 |
| 4 | Archers Post | 0.6375 | 37.6675 | 839 | 0 | RR, T2 | 1 |
| 5 | Ardencaple Farm | 0.0852 | 37.258 | 2271 | 0 | RR | 2 |
| 6 | Castle Forest Station | -0.4083 | 37.3107 | 1927 | 0 | RR | 5 |
| 7 | El Karama | 0.1952 | 36.9038 | 1781 | 0 | RR | 95 |
| 8 | Embori Farm | 0.0677 | 37.3482 | 2691 | 0 | RR | 12 |
| 9 | Embu Met Station | -0.5047 | 37.4579 | 1743 | 0 | RR | 14 |
| 10 | Gathiuru Forest Station | -0.1018 | 37.1159 | 2333 | 0 | RR | 17 |
| 11 | Hombe Forest Station | -0.3508 | 37.1158 | 2017 | 0 | RR | 20 |
| 12 | Jacobson Farm | -0.0432 | 37.0444 | 1913 | 0 | RR | 23 |
| 13 | Kabaru Forest Station | -0.2814 | 37.1535 | 2279 | 0 | RR | 25 |
| 14 | Kisima Farm | 0.1118 | 37.4181 | 2465 | 0 | RR | 35 |
| 15 | Loldaiga Farm | 0.2117 | 37.1219 | 2135 | 0 | RR | 34 |
| 16 | Loruku Farm | -0.0136 | 37.0839 | 1896 | 0 | RR | 38 |
| 17 | Meru Forest Station | 0.0557 | 37.6277 | 1737 | 0 | RR | 45 |
| 18 | Mogwoni Ranch | 0.2284 | 36.9862 | 1683 | 0 | RR | 47 |
| 19 | Mpala Farm | 0.3227 | 36.9038 | 1844 | 0 | RR | 48 |
| 20 | Naro Moru Gate Station | -0.1744 | 37.148 | 2471 | 0 | RR | 61 |
| 21 | Naro Moru Met Station | -0.1704 | 37.214 | 3048 | 0 | RR, T2 | 62 |
| 22 | Nicholson Farm | -0.0886 | 37.0259 | 1916 | 0 | RR | 66 |
| 23 | Nyeri Mow | -0.4162 | 36.9489 | 1854 | 92 | RR | 67 |
| 24 | Ol Donyo Farm | 0.0938 | 37.2929 | 2375 | 0 | RR | 69 |
| 25 | Ontulili Forest Station | 0.0206 | 37.1723 | 2056 | 0 | RR | 75 |
| 26 | Satima Farm | -0.1475 | 37.0101 | 1944 | 0 | RR | 82 |
| 27 | Solio Ranch | -0.2493 | 36.8797 | 1943 | 0 | RR | 87 |
| 28 | Tharua Farm | -0.1046 | 36.8985 | 1865 | 0 | RR | 92 |
| 29 | Kalalu | 0.0817 | 37.1638 | 2027 | 0 | T2 | - |
| 30 | Munyaka | -0.1833 | 37.0596 | 2048 | 0 | T2 | - |