# Peer review of "Sensitivity of precipitation and temperature over Mount Kenya area to physics parameterization options in a high-resolution model simulation performed with WRFV3.8.1"

_Geoscientific Model Development, 2020_

## Short Comment (SC1) · 14 Nov 2020

Dear authors,

even so WRF is a well known model, please add the information how to access the exact WRF code version used for your publication in a "Code availability" section to your article.

Yours,

Astrid Kerkweg (GMD executive Editor)

---

## Author Comment (AC1) · 17 Nov 2020

Dear Astrid Kerkweg,

As suggested, we will include the details about where to find and download the WRF version that we use in our study around Mount Kenya in the new version of the manuscript.

In the code and data availability section we added the following sen-tence: "The Weather Research and Forecasting (WRF) model V3.8.1

is freely available online and can be downloaded from the users' page: https://www2.mmm.ucar.edu/wrf/users/download/get\_sources.html"

Kind regards,

Martina Messmer on behalf of the coauthors

---

## Referee Comment (RC1) · Anonymous Referee #1 · 11 Dec 2020

Summary

The study "Sensitivity of precipitation and temperature over Month Kenia area to physics parameterization option in a high-resolution model simulation performed with WRFV3.8.1" by Messmer et al. submitted to GCM focuses on the analysis of several sensitivity experiments with the Weather Research and Forecasting model version 3.8.1 for Mont Kenia for the year 2008. This work analyzes different parameterization options as well as different nesting strategies (number of nests and nesting ratios). The evaluation of the model performance is carried out in terms of precipitation and tem-

perature by comparing the outputs from the WRF model with observational products from different gridded products but also using observational station datasets.

General comments:

The manuscript is very well written, making it easy to follow. The structure is appropriate with a complete description of the methodology and an adequate list of references. This kind of study is essential before completing climate simulations, being the area of study of high interest as it is a topographically complex region poorly study until now that requires high-resolution climate information to be properly described. In my opinion, the results found in relation to the nesting strategies are very interesting, this being a relevant aspect for properly configuring climate simulations at high resolution. Also, the method used to evaluate the model configurations by comparing the outputs of the model with different observational products seems to be adequate. However, there are several major aspects of the manuscript that should be clarified before its publication.

My major concerns are mainly related to two aspects. The first one is the period used to analyze the WRF model performance for the different configuration options. Being the final goal of the study the selection of a "good configuration" to use WRF for climate runs, why did the authors select a 1-year period (the year 2008) to carry out this study? Did the authors test the model performance in other years with different precipitation characteristics? In my opinion, further analyses could be needed in order to corroborate the results from 2008. To do this, analyses for an additional year, for example, a year considered to be a wet year (as 2006) could be carried out.

On the other hand, I am a little bit confused with the "no cumulus" experiment setup. If I understood well, the authors here have used the WRF model with the cumulus parameterization switch off in all domains (from the 27km_D04/25KM_d03 to the finer domain of 1km of spatial resolution), but, did the coarser domains (i.e., 27km and 25km) also run without convection scheme? I think that more information is required in this regard. Also, why for this parameterization configuration option are there three of

the nesting schemes (i.e., 27kmD04, 25kmD03, 9kmD03) instead of the four used in the other cases (i.e., 27kmD04, 25kmD03, 9kmD03, 25kmD02)?

Additionally, I think that some analysis concerning the ability of the model at a sub-daily scale would be nice to clarify if the "no cumulus" option provides an added value in relation to the other options used with convection physic schemes at this temporal scale.

Specific comments:

L248-250: Please, indicate the method used to interpolate the gridded products. Figure 5: The colors selected for "cumulus3 1-way", "no cumulus", and "ERA5" are hard of difference sometimes, so I would suggest using an additional way (e.g., dotted lines for observations) to clearly show what data are represented in each case. Figure 6 onward: In order to clearly show what option is better, I would suggest adding the correlation patterns between CHIRPS and the different parameterization options, for example, at the bottom of each figure.

Technical corrections:

L115: Velasquez et al. 2020? L178: Please, move the Climatic Research Unit (CRU) definition from L215 to L178. It is the first time that CRU is named.

---

## Referee Comment (RC2) · Anonymous Referee #2 · 21 Dec 2020

General comments:

Messmer et al. present sensitivity simulations with a convection-permitting configuration of the atmospheric model WRF over Mt Kenya. The authors evaluate the impact of several parameterization collections and nest configurations (total number, ratios and interactions), with a focus on monthly total precipitation and monthly mean near-surface air temperature, for a study period of the year 2008. The presented work is intended to determine an optimal configuration for climate simulations over this region. Applying WRF at kilometer-scale grid spacing over Mt Kenya is a novel contribution,

and an improved understanding of current and future variability in precipitation and water resources in this region is of high societal relevance. However, I have a number of concerns about the presented simulations and the analysis that need to be addressed before I can support the publication of the manuscript.

Regarding the simulations:

1. The authors are using WRF v3.8.1, a version of the model that is four years old. The current version is 4.2.1 and numerous code improvements and bug fixes have been issued in the meantime. A brief look at the reported changes shows a major update to the Grell-Freitas scheme and bug fixes for Noah-MP (v.3.9; https://www2.mmm.ucar.edu/wrf/users/wrfv3.9/updates-3.9.html) as well as updates to WSM6 (v. 4.1; https://github.com/wrf-model/WRF/releases). Although ongoing code developments are relevant for all published WRF studies, the older model version employed here may limit the current applicability of the presented results. I suggest that the authors perform an additional simulation of their best-performing configuration (identified as Cumulus3 1-way) with the latest version of WRF to assess the potential impact of model version on their conclusions.

2. It appears that all of the configurations with cumulus parameterizations (CPs) used these schemes in the 3- and 1-km grid spacing domains. Conversely, the No Cumulus configuration did not use a CP in any domain, including the 27- and 25-km grid spacing ones. The established approach is to explicitly resolve convection when grid spacings are less than a few kilometers (e.g., Weisman et al., 1997). Although recent work shows that it might be appropriate to neglect CPs at coarser grid spacings than previously thought (Vergara-Temprado et al., 2020), excluding a CP in D4 could mean that convective processes are inadequately resolved (as the authors propose at line 381). Overall, the authors need to provide a better justification for the CP settings in their tested configurations and, ideally, perform sensitivity simulations to illustrate the impact of (not) using a CP at 3- & 1-km (27- & 25-km) grid spacing.

3. The finest resolution domain in the four-domain configuration is placed upstream of the main regional circulation systems, close to the lateral boundaries of its parent domain. It also appears to be more limited in extent than in the three-domain configuration. Both of these differences will impact the development of small-scale features. Furthermore, the experimental set up does not isolate the impact of the number of nests from the effects of the 1-km domain extent and proximity to boundaries.

A few additional comments on the numerical simulations are provided in the specific comments below.

Regarding the manuscript:

4. The introduction is missing some literature (see specific comments). The analysis would benefit from considering higher temporal resolution data (only monthly totals or means are examined) as well as an enhanced focus on process understanding to provide more insight into the differences between configurations. For example, the differences in near-surface air temperature are attributed to precipitation differences without presenting any supporting analysis of, for example, simulated soil moisture, the latent heat flux, cloudiness, or net radiation. Finally, there is minimal discussion, including of any caveats that might impact the reliability of the results and conclusions (e.g., the sparsity of observations to the southeast of Mt Kenya and above 3048m, the choice of simulation year, etc).

Specific comments:

1. Line 27-28: The discussion of the impact of the Walker circulation changes on interannual precipitation variability should cite at least one of Stefan Hastenrath's papers on this topic (for example, Hastenrath and Polzin, 2004, 2005).

2. Line 38: the long rains are also associated with flooding and drought events (e.g., Kilavi et al., 2018).

3. Lines 39-42: the introduction makes no mention of the Indian Ocean Zonal Mode

and its significant impact on moisture variability in East Africa (e.g., Behera et al., 2006; Nicholson, 2015; Saji et al., 1999; Ummenhofer et al., 2009).

4. Lines 46-48: Wainwright et al., (2019) is relevant to the discussion of the long rains. Please clarify what is meant by "downward trend".

5. Line 55: Are the authors referring specifically to climate simulations?

6. Line 70: Collier et al., (2018) also performed a decadal simulation with WRF at convection-permitting resolutions in East Africa.

7. Line 71: The authors should clarify in this sentence already that it is not only the scale but the ability to neglect a cumulus parameterization that has an impact.

8. Line 107: To be pedantic, WRF is an atmospheric model that can be used for many applications, including regional climate modelling. Please rephrase.

9. Section 2.1: the details provided on the WRF configurations are insufficient to reproduce the study results. Please provide additional details, in Table 1 or elsewhere, including the grid dimensions, selected surface layer scheme, the moisture trigger used with the KF parameterization, diffusion option, and upper boundary condition. Ideally, sample namelists would be made available to interested readers.

10. Line 116: Please provide information on the length and computational expense of the simulations.

11. Line 127: The model top of 50 hPa is low for climate simulations (as per WRF developer recommendations and previous studies), especially over mountainous terrain.

12. Line 130: Please provide the permitted timestep range for each outer grid resolution (or a general relation, e.g., from 3*dx to 8*dx). Also, did the simulations employ restarts? If yes, were they reproducible with the adaptive timestep?

13. Line 145: Whether or not using the lake model improves regional precipitation in the presented simulations could be tested explicitly.

14. Are the authors using the default landuse and terrain datasets as input? How representative are these datasets of conditions on and around Mt. Kenya, including the forest belts and grasslands on the slopes? Mölg and Kaser (2011) reported improved high-elevation simulations with WRF over Kilimanjaro using updated landuse datasets, and both updated landuse and terrain datasets have been employed in recent WRF studies to better represent surface conditions (e.g., Collier et al., 2018, 2019).

15. Line 142: Can the authors also provide which dveg option they are using with Noah-MP? There are issues with certain options for domains containing both hemispheres that could have a significant impact on the results: https://github.com/wrf-model/WRF/issues/707.

16. Line 168: Why were these pressure levels (and not all available levels) between 1000 and 700 hPa selected? Does this choice significantly impact the simulations?

17. Line 170 & Figure 2: the data need to be detrended, if significant trends are present, to examine anomalies. The months of April and May 2008 look very anomalously dry compared with the whole period, and the impact of choosing [only] 2008 as the study period on the results and conclusions needs to be discussed. Also, since the precipitation field in ERA5 is highlighted as being unreliable, would it be preferable to consider CHIRPS data for precipitation in Figure 2?

18. Line 178: Data should be interpolated from higher to lower spatial resolution, as interpolation does not add physically meaningful detail.

19. Section 2.3.3: CHIRPS merges satellite and rain gauge data. Do the authors know if any of the weather station data been assimilated into this product?

20. Lines 241-243: Please move this information to the methods and provide an estimate of statistical significance where applicable.

21. With regards to the statistical methods, the authors do not state which correlation they use and if it is appropriate to have a sample size of 12. In addition, the data have

not been de-seasonalized, which means that a relatively high correlation is to be expected unless WRF fails to capture the seasonal cycle, which should be acknowledged. For their analysis, I suggest that the authors consider a finer temporal resolution, such as pentads, to provide a more detailed and robust assessment of model skill.

22. Lines 327-328: where is this result visible?

23. Lines 331-334: this is the first time that a justification is provided for using an outer domain with a grid spacing of ~25 km. I suggest moving this information to the methods, so the reader understands why the authors include this aspect earlier.

24. Figures 6 to 11: The choice of months could be more robustly justified. In addition, both June and November show low pattern correlations between the observations and the gridded datasets, which undermines using CHIRPS as the main comparison (lines 335—337). Overall, these figures take up a lot of space in the manuscript without providing a great deal of new information. Some suggestions would be to remove the panels for ERA5 and Europe (the authors have already established the poor representation of precipitation) or to replace some map figures with elevational profiles, for compactness and so that the reader can more clearly see some of the reported findings (e.g., lines 403 to 405).

25. Lines 393-396: Please move this information to the methods.

26. Line 459: The authors repeatedly mention that the underestimation of precipitation in the Europe configuration stems from the LW and PBL parameterizations – could they discuss what difference in process representation might be underlying the difference?

27. Line 464-465: Can the authors please clarify what they mean about one-way vs two-way nesting?

Technical comments:

1. Line 129: The sentence "This is, because. . ." is unnecessary, please remove.

[Figure]

2. Line 132: Please change "in order to improve" to "to optimize."

3. Line 164 to 167: The forcing variables are well known and documented, so I suggest deleting these sentences.

4. Lines 190 and 201: It is clear that data are only compared for the study period, please remove.

5. Lines 234 to 237 are repetitive and could be removed.

6. Line 341: CHIRPS is not a model, please rephrase.

7. Line 344: "Bearing"

8. Line 436-437: Please clarify that these parameterizations are not varied independently.

Figures:

Figure 1: I suggest adding the weather station locations to the plots of D3/D4, as the reader does not see where they are located before encountering Figure 6. Also, is Figure 1 referenced in the text?

References:

Behera, S. K., Luo, J. J., Masson, S., Delecluse, P., Gualdi, S., Navarra, A. and Yamagata, T.: Impact of the Indian Ocean Dipole on the East African Short Rains: A CGCM Study* Swadhin, J. Clim., 19(7), 1361, doi:10.1175/JCLI9018.1, 2006.

Collier, E., Mölg, T. and Sauter, T.: Recent atmospheric variability at Kibo summit, Kilimanjaro, and its relation to climate mode activity, J. Clim., 31(10), 3875–3891, doi:10.1175/JCLI-D-17-0551.1, 2018.

Collier, E., Sauter, T., Mölg, T. and Hardy, D.: The influence of tropical cyclones on circulation, moisture transport, and snow accumulation at Kilimanjaro during the 2006 - 2007 season, J. Geophys. Res. Atmos., 124(13), 6919–6928,

doi:10.1029/2019JD030682, 2019.

Hastenrath, S. and Polzin, D.: Exploring the predictability of the "short rains" at the coast of East Africa, Int. J. Climatol., 2004.

Hastenrath, S. and Polzin, D.: Mechanisms of climate anomalies in the equatorial Indian Ocean, J. Geophys. Res. Earth Surf., 110(D8), D08113, 2005.

Kilavi, M., MacLeod, D., Ambani, M., Robbins, J., Dankers, R., Graham, R., Helen, T., Salih, A. A. M. and Todd, M. C.: Extreme rainfall and flooding over Central Kenya Including Nairobi City during the long-rains season 2018: Causes, predictability, and potential for early warning and actions, Atmosphere (Basel)., 9(12), 472, doi:10.3390/atmos9120472, 2018.

Mölg, T. and Kaser, G.: A new approach to resolving climate-cryosphere relations: Downscaling climate dynamics to glacier-scale mass and energy balance without statistical scale linking, J. Geophys. Res. Atmos., 116(16), doi:10.1029/2011JD015669, 2011.

Nicholson, S. E.: Long-term variability of the East African 'short rains' and its links to large-scale factors, Int. J. Climatol., 35(13), 3979–3990, 2015.

Saji, N. H., Goswami, B. N. and Vinayachandran, P. N.: A dipole mode in the tropical Indian Ocean, Nature, 401, 360–363, 1999.

Ummenhofer, C. C., Sen Gupta, A. and England, M. H.: Contributions of Indian Ocean sea surface temperatures to enhanced East African rainfall, J. Clim., 22, 993–1013, doi:10.1175/2008JCLI2493.1, 2009.

Vergara-Temprado, J., Ban, N., Panosetti, D., Schlemmer, L. and Schär, C.: Climate models permit convection at much coarser resolutions than previously considered, J. Clim., doi:10.1175/JCLI-D-19-0286.1, 2020.

Wainwright, C. M., Marsham, J. H., Keane, R. J., Rowell, D. P., Finney, D. L., Black,

E. and Allan, R. P.: 'Eastern African Paradox' rainfall decline due to shorter not less intense Long Rains, npj Clim. Atmos. Sci., doi:10.1038/s41612-019-0091-7, 2019.

Weisman, M. L., Skamarock, W. C. and Klemp, J. B.: The Resolution Dependence of Explicitly Modeled Convective Systems, Mon. Weather Rev., 125(4), 527–548, 1997.

---

## Author Comment (AC2) · 13 Jan 2021

**Reply to Anonymous Referee #1**

*Summary*

*The study "Sensitivity of precipitation and temperature over Month Kenia area to physics parameterization option in a high-resolution model simulation performed with WRFV3.8.1" by Messmer et al. submitted to GCM focuses on the analysis of several sensitivity experiments with the Weather Research and Forecasting model version3.8.1 for Mont Kenia for the year 2008. This work analyzes different parameterization options as well as different nesting strategies (number of nests and nesting ratios). The evaluation of the model performance is carried out in terms of precipitation and temperature by comparing the outputs from the WRF model with observational products from different gridded products but also using observational station datasets.*

*General comments:*

*The manuscript is very well written, making it easy to follow. The structure is appropriate with a complete description of the methodology and an adequate list of references. This kind of study is essential before completing climate simulations, being the area of study of high interest as it is a topographically complex region poorly study until now that requires high-resolution climate information to be properly described. In my opinion, the results found in relation to the nesting strategies are very interesting, this being a relevant aspect for properly configuring climate simulations at high resolution. Also, the method used to evaluate the model configurations by comparing the outputs of the model with different observational products seems to be adequate. However, there are several major aspects of the manuscript that should be clarified before its publication.*

Thank you for reading our manuscript so carefully and for taking the time to review it. Thank you for asking critical questions, which help us to improve the quality of the paper.

*My major concerns are mainly related to two aspects. The first one is the period used to analyze the WRF model performance for the different configuration options. Being the final goal of the study the selection of a "good configuration" to use WRF for climate runs, why did the authors select a 1-year period (the year 2008) to carry out this study? Did the authors test the model performance in other years with different precipitation characteristics? In my opinion, further analyses could be needed in order to corroborate the results from 2008. To do this, analyses for an additional year, for example, a year considered to be a wet year (as 2006) could be carried out.*

We have decided to choose one year, as the selected domain setups are computationally quite heavy (38'000 CPU hours per model year and 7 TB of storage space), so we decided to run a large set of experiments, at the expense of a somewhat shorter simulation. We have tested 4 different domain setups and at least 5 parameterization options for each setup, which results in more than 20 years of simulation. This corresponds almost to a full climatology. Owing to the fact that the resolution is very high, longer and more simulations can only be afforded when reducing the number of parameterization options. Nevertheless, we agree that it would be good to have an additional check using a wetter year as well. Hence, we have decided to start one simulation with the year 2006 and the optimal setting for the year 2008.

*On the other hand, I am a little bit confused with the "no cumulus" experiment setup. If I understood well, the authors here have used the WRF model with the cumulus parameterization switch off in all domains (from the 27km_D04/25KM_d03 to the finer domain of 1km of spatial resolution), but did the coarser domains (i.e., 27km and25km) also run without convection scheme? I think that more information is required in this regard. Also, why for this parameterization configuration option are there three of the nesting schemes (i.e., 27kmD04, 25kmD03, 9kmD03) instead of the four used in the other cases (i.e., 27kmD04, 25kmD03, 9kmD03, 25kmD02)?*

Thank you for pointing out that the description of the "No cumulus" parameterization option is still unclear. For the "No cumulus" simulation we have turned off the cumulus parameterization in all domains, i.e., we have turned it off also in the 27/25 km and 9 km domains, where it is normally suggested to use a cumulus parameterization. Note further that the cumulus scheme is turned off for grid spacings equal or finer than 5 km in all the experiments.

This is also why there is no "No cumulus" parameterization experiment for 5km_D02 setting. As the same parameterizations are used in both "No cumulus" and "Cumulus 3-1Way" except for the cumulus scheme, these two experiments are identical for the 5km_D02 setting since cumulus parameterization is switched off in both setups at that resolution. We understand that the text in the manuscript still leads to some confusion and that is why we will be more precise in describing this experiment in the next version of the manuscript.

*Additionally, I think that some analysis concerning the ability of the model at a sub-daily scale would be nice to clarify if the "no cumulus" option provides an added value in relation to the other options used with convection physic schemes at this temporal scale.*

This is correct, a sub-daily analysis would be nice to further investigate the skill of each parameterization and in particular for the "No cumulus" parameterization this would be interesting. Since most of the observations are only available on a daily basis, such an analysis is a bit difficult, but we will try to include a sub-daily analysis based on IMERG, as we have 3-hourly data available there.

*Specific comments:*

*L248-250: Please, indicate the method used to interpolate the gridded products.*

Bilinear interpolation was applied to the gridded products. This information will be included in the new version of the manuscript.

*Figure5: The colors selected for "cumulus3 1-way", "no cumulus", and "ERA5" are hard of difference sometimes, so I would suggest using an additional way (e.g., dotted lines for observations) to clearly show what data are represented in each case.*

It is a valid point to add dotted lines for the observations, so we will adapt this figure accordingly.

*Figure 6onward: In order to clearly show what option is better, I would suggest adding the correlation patterns between CHIRPS and the different parameterization options, for example, at the bottom of each figure.*

Thank you for this suggestion. This is a good idea, and we will include this number as suggested in the next version of the manuscript.

*Technical corrections:*

*L115: Velasquez et al. 2020?*

*L178: Please, move the Climatic Research Unit (CRU) definition from L215 to L178. It is the first time that CRU is named.*

We will address these two technical corrections in the new version of the manuscript as suggested by the reviewer.

---

## Author Comment (AC3) · 13 Jan 2021

Reply to Anonymous Referee #2

General comments:

*Messmer et al. present sensitivity simulations with a convection-permitting configuration of the atmospheric model WRF over Mt Kenya. The authors evaluate the impact of several parameterization collections and nest configurations (total number, ratios and interactions), with a focus on monthly total precipitation and monthly mean near-surface air temperature, for a study period of the year 2008. The presented work is intended to determine an optimal configuration for climate simulations over this region. Applying WRF at kilometer-scale grid spacing over Mt Kenya is a novel contribution, and an improved understanding of current and future variability in precipitation and water resources in this region is of high societal relevance. However, I have a number of concerns about the presented simulations and the analysis that need to be addressed before I can support the publication of the manuscript.*

Thank you for going through our manuscript so carefully and for taking the time to review it. Thank you for providing insightful comments which improve the quality of our paper.

Regarding the simulations:

1. *The authors are using WRF V3.8.1, a version of the model that is four years old. The current version is 4.2.1 and numerous code improvements and bug fixes have been issued in the meantime. A brief look at the reported changes shows a major update to the Grell-Freitas scheme and bug fixes for Noah-MP (V3.9; https://www2.mmm.ucar.edu/wrf/users/wrfv3.9/updates-3.9.html) as well as updates to WSM6 (V4.1; https://github.com/wrf-model/WRF/releases). Although ongoing code developments are relevant for all published WRF studies, the older model version employed here may limit the current applicability of the presented results. I suggest that the authors perform an additional simulation of their best-performing configuration (identified as Cumulus3 1-way) with the latest version of WRF to assess the potential impact of model version on their conclusions.*

   The used version V3.8.1 is a stable and widely used version of WRF. Since we have started first simulations more than two years ago, we have decided to stick to the same version, in order to keep some level of consistency throughout the course of the experiment. Additionally, since the model is updated quite frequently (every year there is a major update, while there are new subversions on an approximately 1 to 3-monthly basis), it is unusual to work with the most up to date version. Especially because this version is also often subject to errors and instability.

   We have, nevertheless, started a simulation with the optimal setting using the most recent WRF version 4.2.1. The currently used WRF version 3.8.1 was compiled with Intel, but as the cumulus parameterization 3 (Grell-Freitas) is not running with an optimized compilation on our high-performance computer, we had to use a different compiler. With PGI, WRF V4.2.1 is now running with our optimal setup, but it must be kept in mind that small changes might be introduced because of differences in the numeric solution of WRF when using another compiler. We will analyse the results of the simulation and decide if it is worthwhile to include additional figures or if a short notice is sufficient in the next version of the manuscript.

2. *It appears that all of the configurations with cumulus parameterizations (CPs) used these schemes in the 3- and 1-km grid spacing domains. Conversely, the No Cumulus configuration did not use a CP in any domain, including the 27- and 25-km grid spacing ones. The established approach is to explicitly resolve convection when grid spacings are less than a few kilometers (e.g., Weisman et al., 1997). Although recent work shows that it might be appropriate to neglect CPs at coarser grid spacings than previously thought (Vergara-Temprado et al., 2020), excluding a CP in D4 could mean that convective processes are inadequately resolved (as the authors propose at line381). Overall, the*

*authors need to provide a better justification for the CP settings in their tested configurations and, ideally, perform sensitivity simulations to illustrate the impact of (not) using a CP at 3- & 1-km (27- & 25-km) grid spacing.*

It seems that there is some misunderstanding, and we have realized that Table 1 in our manuscript is not precise enough. Hence, we will update Table 1 in the new version of the manuscript, so that it is clear that we are not using CP in domains that have a grid spacing of 5 km or less. This scale is certainly convection permitting and hence, no parameterization is needed here, as the reviewer points out correctly. For the larger domains it is generally suggested to use a cumulus parameterization and here we already perform various experiments with different schemes and also turning it off as suggested by Vergara-Temprado et al. (2020). Since we are not using a cumulus scheme in the fine domains, and because we have done several experiments with turning on or off the CP in the coarser domains, there is no need to perform additional sensitivity experiments.

Nevertheless, we will include a clearer motivation, why we turn off or make use of a cumulus parameterization in the next version of the manuscript.

3. *The finest resolution domain in the four-domain configuration is placed upstream of the main regional circulation systems, close to the lateral boundaries of its parent domain. It also appears to be more limited in extent than in the three-domain configuration. Both of these differences will impact the development of small-scale features. Furthermore, the experimental set up does not isolate the impact of the number of nests from the effects of the 1-km domain extent and proximity to boundaries.*

The domain is more than 50 grid points away from the east boundary of D03 and more than 30 grid points from the northern edge of D03. This is much further away than the 5 to 10 grid points around the boundaries that are needed for the relaxation of the outer boundary conditions (Rummukainen, 2010). Hence, we do not expect any problems from the boundaries. Furthermore, as the grid spacing is already very fine, it is expected that the two domains have a rather similar solution as several processes can be explicitly described. We will provide a more zoomed in figure for the reviewer in the point-to-point response, as in Fig. 1 of the manuscript the domain borders are rather thick (and well visible), so that it seems that the domains are very close to each other.

Rummukainen, M. (2010), State-of-the-art with regional climate models. WIREs Clim Change, 1: 82-96. https://doi.org/10.1002/wcc.8

*A few additional comments on the numerical simulations are provided in the specific comments below.*

*Regarding the manuscript:*

4. *The introduction is missing some literature (see specific comments). The analysis would benefit from considering higher temporal resolution data (only monthly totals or means are examined) as well as an enhanced focus on process understanding to provide more insight into the differences between configurations. For example, the differences in near-surface air temperature are attributed to precipitation differences without presenting any supporting analysis of, for example, simulated soil moisture, the latent heat flux, cloudiness, or net radiation. Finally, there is minimal discussion, including of any caveats that might impact the reliability of the results and conclusions (e.g., the sparsity of observations to the southeast of Mt Kenya and above 3048m, the choice of simulation year, etc).*

We will expand the introduction with the suggested literature. Since reviewer 1 also requests a sub-daily analysis, we will try to include such an analysis, even if we are only able to perform this with IMERG, as all the other datasets and observations are available on a daily basis only. If this does not provide insightful results because of the restriction to IMERG, we can further investigate daily values or pentads.

We will further investigate the link between temperature and precipitation, by investigating LH, soil moisture, clouds etc as suggested by the reviewer. We will decide afterwards, if we will include further figures directly into the manuscript or the supplementary, or if a short notice in the text is sufficient.

Further, we will include a more extended discussion on the reliability of the data in the next version of the manuscript.

Specific comments:

1. *Line 27-28: The discussion of the impact of the Walker circulation changes on in-terannual precipitation variability should cite at least one of Stefan Hastenrath's papers on this topic (for example, Hastenrath and Polzin, 2004, 2005).*

   Thank you for pointing this out, we will include this in the next version of the manuscript.

2. *Line 38: the long rains are also associated with flooding and drought events (e.g., Kilavi et al., 2018).*

   We will be more precise with this statement in the next version of the manuscript.

3. *Lines 39-42: the introduction makes no mention of the Indian Ocean Zonal Mode and its significant impact on moisture variability in East Africa (e.g., Behera et al., 2006; Nicholson, 2015; Saji et al., 1999; Ummenhofer et al., 2009).*

   Thank you for pointing this out, it will be included in the next version of the manuscript.

4. *Lines 46-48: Wainwright et al., (2019) is relevant to the discussion of the long rains. Please, clarify what is meant by "downward trend".*

   We will clarify this in the next version of the manuscript.

5. *Line 55: Are the authors referring specifically to climate simulations?*

   Yes, we do. We will be more specific here in the next version of the manuscript.

6. *Line 70: Collier et al., (2018) also performed a decadal simulation with WRF at convection-permitting resolutions in East Africa.*

   We will include this literature in our introduction as well.

7. *Line 71: The authors should clarify in this sentence already that it is not only the scale but the ability to neglect a cumulus parameterization that has an impact.*

   We will make it clear in the new version of the manuscript.

8. *Line 107: To be pedantic, WRF is an atmospheric model that can be used for many applications, including regional climate modelling. Please rephrase.*

   We will rephrase it to: We adopt the numerical weather prediction model WRF …

9. *Section 2.1: the details provided on the WRF configurations are insufficient to re-produce the study results. Please provide additional details, in Table 1 or elsewhere, including the grid dimensions, selected surface layer scheme, the moisture trigger used with the KF parameterization, diffusion option, and upper boundary condition. Ideally, sample namelists would be made available to interested readers.*

   We will add our namelists to Zenodo, so that they are available for anybody.

10. *Line 116: Please provide information on the length and computational expense of the simulations.*

    All the experiments used up around 750'000 CPU hours and they fill up around 140 TB of storage space.

11. Line 127: The model top of 50 hPa is low for climate simulations (as per WRF developer recommendations and previous studies), especially over mountainous terrain.

50 hPa is the standard best practice p_top value for the WRF simulation (https://www2.mmm.ucar.edu/wrf/users/namelist_best_prac_wrf.html). Additionally, it is one of the most widely used value also in recent studies over complex terrain, e.g., over Europe (e.g., Dörenkämper et al., 2020) and this level also depends on the available levels from the input data, i.e., ERA5 or CESM for our later studies.

Dörenkämper, M., Olsen, B. T., Witha, B., Hahmann, A. N., Davis, N. N., Barcons, J., ... & Mann, J. (2020). The Making of the New European Wind Atlas–Part 2: Production and Evaluation. *Geoscientific Model Development*, *13*(10), 5079-5102. https://doi.org/10.5194/gmd-13-5079-2020

12. *Line 130: Please provide the permitted timestep range for each outer grid resolution (or a general relation, e.g., from 3\* to 8\*dx). Also, did the simulations employ restarts? If yes, were they reproducible with the adaptive timestep?*

The range of the dt for the outermost domains is given in line 129-131 in the current version of the manuscript. If the reader needs more details about the exact setting, she/he can refer to the namelist on Zenodo, which we will upload before the next submission.

We are using restart files and we will test if the restart files are reproducible for the upcoming revision. The result of this test will be added as a short discussion to the next version of the manuscript.

13. *Line 145: Whether or not using the lake model improves regional precipitation in the presented simulations could be tested explicitly.*

This is correct. We have already started a new simulation, where the lake model is turned off. We will add the results to the next version of the manuscript.

14. *Are the authors using the default land use and terrain datasets as input? How representative are these datasets of conditions on and around Mt. Kenya, including the forest belts and grasslands on the slopes? Mölg and Kaser (2011) reported improved high-elevation simulations with WRF over Kilimanjaro using updated land use datasets, and both updated land use and terrain datasets have been employed in recent WRF studies to better represent surface conditions (e.g., Collier et al., 2018, 2019).*

It is true that there are other published studies where updated land use data is used. However, we decided to use the default data provided within the model, because in a next step of our study, we plan to run different climatologies for the region with different land use options. In order to identify the effect of changed land use, we first need a base state to compare it to.

15. *Line 142: Can the authors also provide which dveg option they are using with Noah-MP? There are issues with certain options for domains containing both hemispheres that could have a significant impact on the results: https://github.com/wrf-model/WRF/issues/707.*

We have used the default option and we are also aware of the problem along the equator. However, when we started the first experiments, no solution for this issue was available except changing to another land surface model, something that we did not want to do. We have started a new simulation with the optimal setting and option dveg = 9, to estimate how significant the impact on the results of the simulation actually is.

16. *Line 168: Why were these pressure levels (and not all available levels) between1000 and 700 hPa selected? Does this choice significantly impact the simulations?*

The plan is to use this model setup with input from a climate model, where we also have to interpolate the levels and it is not meaningful to just interpolate various levels from a limited number of sigma-pressure levels. Our intention is to have the model run as freely as possible, as we will need this for the climate simulations. Also, other climate models from, e.g., CMIP5 and 6 do not provide this many pressure levels. Hence, the selected levels are sufficient to drive a regional climate model.

17. *Line 170 & Figure 2: the data need to be detrended, if significant trends are present, to examine anomalies. The months of April and May 2008 look very anomalously dry compared with the whole period, and the impact of choosing [only] 2008 as the study period on the results and conclusions needs to be discussed. Also, since the precipitation field in ERA5 is highlighted as being unreliable, would it be preferable to consider CHIRPS data for precipitation in Figure 2?*

    Thank you for this comment. We will detrend the data and redo the figure, also with CHIRPS to see how this changes the results. Furthermore, a more detailed discussion on the selected year will be added.

18. Line 178: Data should be interpolated from higher to lower spatial resolution, as interpolation does not add physically meaningful detail.

    It is correct that no detail is added to the coarser resolution if we interpolate from the lower to the higher resolution. However, if we interpolate from lower to higher resolution, we lose the fine scale information of WRF, while the other way round we do not change the physical detail. Additionally, the grid of WRF is the common thread, while all the gridded data sets have different grid resolutions. Furthermore, we expect that if we interpolate the WRF grid points to a coarser grid, the spatial correlations would be higher, and the test is less restrictive. For these reasons we prefer to keep it the way it is right now. We will add a justification for this approach in the next version of the manuscript.

19. *Section 2.3.3: CHIRPS merges satellite and rain gauge data. Do the authors know if any of the weather station data been assimilated into this product?*

    As far as we know the observations gathered by CETRAD are independent from any other used station data.

20. *Lines 241-243: Please move this information to the methods and provide an estimate of statistical significance where applicable.*

    We will consider moving those lines to Section 2.1 as suggested, but we are not sure if that change improves the readability of the paper. The statistical significance of the results was not included in the current version as only 12 monthly values were used to calculate the temporal correlations. If the (sub-)daily analysis is performed in the next version, the statistical significance will be evaluated and included.

21. *With regards to the statistical methods, the authors do not state which correlation they use and if it is appropriate to have a sample size of 12. In addition, the data have not been de-seasonalized, which means that a relatively high correlation is to be expected unless WRF fails to capture the seasonal cycle, which should be acknowledged. For their analysis, I suggest that the authors consider a finer temporal resolution, such as pentads, to provide a more detailed and robust assessment of model skill.*

    As we said before, also reviewer 1 suggests to perform the same analysis but with a finer temporal resolution. This will be evaluated in the next steps. Additionally, we think that a deseasonalization with only one year is not possible, as you would need a longer period to capture a typical seasonal cycle. We will include a short note on this fact in the new version of the manuscript.

22. *Lines 327-328: where is this result visible?*

    This result is observable in the insets of Figure 5. There, it can be seen that the annual precipitation sums for the Cumulus 3 options is below the one from ERA5. We will add a label towards Figure 5 in the text, to make it clearer.

23. *Lines 331-334: this is the first time that a justification is provided for using an outer domain with a grid spacing of ~25 km. I suggest moving this information to the methods, so the reader understands why the authors include this aspect earlier.*

    This is a valid point. We will do so in the next version of the manuscript.

*24. Figures 6 to 11: The choice of months could be more robustly justified. In addition, both June and November show low pattern correlations between the observations and the gridded datasets, which undermines using CHIRPS as the main comparison (lines 335-337). Overall, these figures take up a lot of space in the manuscript without providing a great deal of new information. Some suggestions would be to remove the panels for ERA5 and Europe (the authors have already established the poor representation of precipitation) or to replace some map figures with elevational profiles, for compactness and so that the reader can more clearly see some of the reported findings (e.g., lines 403 to 405).*

We will consider your suggestions regarding those figures and the selection of the months.

*25. Lines 393-396: Please move this information to the methods.*

We will move this information to Section 2.3.5 as suggested.

*26. Line 459: The authors repeatedly mention that the underestimation of precipitation in the Europe configuration stems from the LW and PBL parameterizations – could they discuss what difference in process representation might be underlying the difference?*

We will try to include more information regarding the differences in the next version of the manuscript.

*27. Line 464-465: Can the authors please clarify what they mean about one-way vs two-way nesting?*

In the two-way nesting option, the results from the nest overwrite parent domain results. In the one-way nesting, the results from the nests do not overwrite the data in the parent domain and independent results are obtained in each domain. We will make this difference clear in the new version of the manuscript.

*Technical comments:*

*1. Line 129: The sentence "This is, because..." is unnecessary, please remove.*

*2. Line 132: Please change "in order to improve" to "to optimize."*

*3. Line 164 to 167: The forcing variables are well known and documented, so I suggest deleting these sentences.*

*4. Lines 190 and 201: It is clear that data are only compared for the study period, please remove.*

*5. Lines 234 to 237 are repetitive and could be removed.*

*6. Line 341: CHIRPS is not a model, please rephrase.*

*7. Line 344: "Bearing"*

*8. Line 436-437: Please clarify that these parameterizations are not varied independently.*

We will address all these technical comments in the new version of the manuscript as suggested by the reviewer.

*Figures:*

*Figure 1: I suggest adding the weather station locations to the plots of D3/D4, as the reader does not see where they are located before encountering Figure 6. Also, is Figure 1 referenced in the text?*

We will add the observations to Figure 1 and add the missing reference in the text. Thank you for pointing this out.

References:

Behera, S. K., Luo, J. J., Masson, S., Delecluse, P., Gualdi, S., Navarra, A. and Yamagata, T.: Impact of the Indian Ocean Dipole on the East African Short Rains: A CGCM Study* Swadhin, J. Clim., 19(7), 1361, doi:10.1175/JCLI9018.1, 2006.

Collier, E., Mölg, T. and Sauter, T.: Recent atmospheric variability at Kibo summit, Kilimanjaro, and its relation to climate mode activity, J. Clim., 31(10), 3875–3891, doi:10.1175/JCLI-D-17-0551.1, 2018.

Collier, E., Sauter, T., Mölg, T. and Hardy, D.: The influence of tropical cyclones on circulation, moisture transport, and snow accumulation at Kilimanjaro during the 2006 - 2007 season, J. Geophys. Res. Atmos., 124(13), 6919–6928, doi:10.1029/2019JD030682, 2019.

Hastenrath, S. and Polzin, D.: Exploring the predictability of the "short rains" at the coast of East Africa, Int. J. Climatol., 2004.

Hastenrath, S. and Polzin, D.: Mechanisms of climate anomalies in the equatorial Indian Ocean, J. Geophys. Res. Earth Surf., 110(D8), D08113, 2005.

Kilavi, M., MacLeod, D., Ambani, M., Robbins, J., Dankers, R., Graham, R., Helen, T., Salih, A. A. M. and Todd, M. C.: Extreme rainfall and flooding over Central Kenya Including Nairobi City during the long-rains season 2018: Causes, predictability, and potential for early warning and actions, Atmosphere (Basel)., 9(12),472, doi:10.3390/atmos9120472, 2018.

Mölg, T. and Kaser, G.: A new approach to resolving climate-cryosphere relations: Downscaling climate dynamics to glacier-scale mass and energy balance without sta-tistical scale linking, J. Geophys. Res. Atmos., 116(16), doi:10.1029/2011JD015669, 2011.

Nicholson, S. E.: Long-term variability of the East African 'short rains' and its links to large-scale factors, Int. J. Climatol., 35(13), 3979–3990, 2015.

Saji, N. H., Goswami, B. N. and Vinayachandran, P. N.: A dipole mode in the tropical Indian Ocean, Nature, 401, 360–363, 1999.

Ummenhofer, C. C., Sen Gupta, A. and England, M. H.: Contributions of Indian Ocean sea surface temperatures to enhanced East African rainfall, J. Clim., 22, 993–1013, doi:10.1175/2008JCLI2493.1, 2009.

Vergara-Temprado, J., Ban, N., Panosetti, D., Schlemmer, L. and Schär, C.: Climate models permit convection at much coarser resolutions than previously considered, J.Clim., doi:10.1175/JCLI-D-19-0286.1, 2020.

Wainwright, C. M., Marsham, J. H., Keane, R. J., Rowell, D. P., Finney, D. L., Black, E. and Allan, R. P.: 'Eastern African Paradox' rainfall decline due to shorter not less intense Long Rains, npj Clim. Atmos. Sci., doi:10.1038/s41612-019-0091-7, 2019.

Weisman, M. L., Skamarock, W. C. and Klemp, J. B.: The Resolution Dependence of Explicitly Modeled Convective Systems, Mon. Weather Rev., 125(4), 527–548, 1997

---

## Author Response (AR1)

**Point-to-point response to anonymous referee #1**

**Summary**

The study "Sensitivity of precipitation and temperature over Month Kenia area to physics parameterization option in a high-resolution model simulation performed with WRFV3.8.1" by Messmer et al. submitted to GCM focuses on the analysis of several sensitivity experiments with the Weather Research and Forecasting model version3.8.1 for Mont Kenia for the year 2008. This work analyzes different parameterization options as well as different nesting strategies (number of nests and nesting ratios). The evaluation of the model performance is carried out in terms of precipitation and temperature by comparing the outputs from the WRF model with observational products from different gridded products but also using observational station datasets.

**General comments:**

The manuscript is very well written, making it easy to follow. The structure is appropriate with a complete description of the methodology and an adequate list of references. This kind of study is essential before completing climate simulations, being the area of study of high interest as it is a topographically complex region poorly study until now that requires high-resolution climate information to be properly described. In my opinion, the results found in relation to the nesting strategies are very interesting, this being a relevant aspect for properly configuring climate simulations at high resolution. Also, the method used to evaluate the model configurations by comparing the outputs of the model with different observational products seems to be adequate. However, there are several major aspects of the manuscript that should be clarified before its publication.

**Thank you for reading our manuscript so carefully and for taking the time to review it. Thank you for asking critical questions, which help us to improve the quality of the paper.**

My major concerns are mainly related to two aspects. The first one is the period used to analyze the WRF model performance for the different configuration options. Being the final goal of the study the selection of a "good configuration" to use WRF for climate runs, why did the authors select a 1-year period (the year 2008) to carry out this study? Did the authors test the model performance in other years with different precipitation characteristics? In my opinion, further analyses could be needed in order to corroborate the results from 2008. To do this, analyses for an additional year, for example, a year considered to be a wet year (as 2006) could be carried out.

We have decided to choose one year, as the selected domain setups are computationally quite heavy (38'000 CPU hours per model year and 7 TB of storage space), so we decided to run a large set of experiments, at the expense of a somewhat shorter simulation. We have tested 4 different domain setups and at least 5 parameterization options for each setup, which results in more than 20 years of simulation. This corresponds almost to a full climatology. Owing to the fact that the resolution is very high, longer and more simulations can only be afforded when reducing the number of parameterization options. Nevertheless, we agree that it is good to have one additional sensitivity test using a wetter year as well.

Following the suggestion of the reviewer, we have analyzed the year 2006, which presents wetter conditions than 2008. In this case, we run the simulation only for the best configuration ("Cumulus3 1-way"), and the results of this experiment are compared to the gridded datasets ERA5, IMERG and CHIRPS, and to the available weather station data for 2006. Overall, we find that the optimal setting applied to 2006 resemble the findings of 2008 with even higher pattern correlation to station data than for 2008.

We added a few paragraphs to the new version of the manuscript related to the results of the year 2006 at the end of Section 3.1 and in the conclusions. Figure R1.1 and Figure R1.2 are added as supplementary material to the manuscript.

On the other hand, I am a little bit confused with the "no cumulus" experiment setup. If I understood well, the authors here have used the WRF model with the cumulus parameterization switch off in all domains (from the 27km\_D04/25KM\_d03 to the finer domain of 1km of spatial resolution), but did the coarser domains (i.e., 27km and25km) also run without convection scheme? I think that more information is required in this regard. Also, why for this parameterization configuration option are there three of the nesting schemes (i.e., 27kmD04, 25kmD03, 9kmD03) instead of the four used in the other cases (i.e., 27kmD04, 25kmD03, 9kmD03, 25kmD02)?

Thank you for pointing out that the description of the "No cumulus" parameterization option is still unclear. For the "No cumulus" simulation we have turned off the cumulus parameterization in all domains, i.e., we have turned it off also in the 27/25 km and 9 km domains, where it is normally suggested to use a cumulus parameterization. Note further that the cumulus scheme is turned off for grid spacings equal or finer than 5 km in all the experiments.

This is also why there is no "No cumulus" parameterization experiment for the 5km\_D02 setting. As the same parameterizations are used in both "No cumulus" and "Cumulus 3 1-way" except for the cumulus scheme, these two experiments are identical for the 5km\_D02 setting since cumulus parameterization is switched off in both setups at that resolution. We understand that the text in the manuscript still leads to some confusion. We have updated Table 1 in the manuscript, as this table was imprecise in the former version of the manuscript. Now, we have removed the name of the cumulus parameterization if all the domains are in the convection permitting scale and we have written the domains in bold letters if a cumulus parameterization is used. Furthermore, we have included the fourth simulation of the "No Cumulus" set up and have added a footnote explaining that this last simulation is the same one as the last in the "Cumulus3 1-way" experiment. We have also added this information to Section 2.1. in the manuscript. We hope that we were able to clarify the misleading statements related to the different experiments for the reader.

Additionally, I think that some analysis concerning the ability of the model at a sub-daily scale would be nice to clarify if the "no cumulus" option provides an added value in relation to the other options used with convection physic schemes at this temporal scale.

This is correct, a sub-daily analysis is nice to further investigate the skill of each parameterization and in particular for the "No cumulus" parameterization such an analysis is interesting. Since most of the observations are only available on a daily basis, a sub-daily analysis is a bit difficult, but we performed it on a monthly basis with IMERG, as we have 1-hourly data available there. The mean hourly precipitation has been calculated over the entire domain (D4 and D2) for each sensitivity experiment and IMERG. Additionally, to ease the interpretation of the results, the local time in Kenya (+2 UTC) has been employed while plotting the data.

Figure R1.3 (left column) shows the daily cycle for D4. All the simulations show a similar daily cycle throughout the year, except for some differences in intensities. All the simulations describe precipitation throughout the day fairly similar, except for the "No Cumulus" experiment, which overestimates precipitation rates in the afternoon in most of the months. Since all the experiments resolve convective processes explicitly it is expected that the "No Cumulus" option does not add any value in D4. This is different if we compare the simulations in domain D2. Here, except for the "No Cumulus" experiment all the simulations have parameterized convective processes. Figure R1.3 (right column) reveals clearly the added value of not using a parameterization scheme, as peak precipitation of IMERG and the "No Cumulus" option agree, while the other options simulate the peak a few hours too early.

The results of domain D2 are very interesting and we have added a small paragraph regarding the timing of peak precipitation in the "No Cumulus" experiment to the results. As we do not show any result for D2 in the manuscript, adding Figure R1.3 to the manuscript would be misleading. We have added this figure to the supplementary material.

Specific comments:

L248-250: Please, indicate the method used to interpolate the gridded products.

We used bilinear interpolation when comparing the gridded products. This information is included in the new version of the manuscript.

We have added the exact interpolation method in Section 2.3 (L179.) as it is the place where the interpolation is first mentioned. We have adapted the sentence to:

"Please note that the gridded data sets are bi-linearly interpolated (Climate Data Operator (CDO; Schulzweida, 2019) to the grid of the WRF domain that it is compared to, and the nearest point to the station is considered afterwards."

Figure 5: The colors selected for "Cumulus3 1-way", "No Cumulus", and "ERA5" are hard of difference sometimes, so I would suggest using an additional way (e.g., dotted lines for observations) to clearly show what data are represented in each case.

It is a valid point to add dotted lines for the observations, so we have adapted this figure accordingly.

Figure 6 onward: In order to clearly show what option is better, I would suggest adding the correlation patterns between CHIRPS and the different parameterization options, for example, at the bottom of each figure.

Thank you for this suggestion. This is a good idea, and we have included this number in the lower right corner of each panel. Apart from the pattern correlation against CHIRPS, we have decided to include the pattern correlation against the weather station data in each plot as well.

Technical corrections:

L115: Velasquez et al. 2020?

We addressed this technical correction in the new version of the manuscript as suggested by the reviewer.

L178: Please, move the Climatic Research Unit (CRU) definition from L215 to L178. It is the first time that CRU is named.

We have deleted the section on CRU, because Fig. 9-11 are adapted in accordance with a suggestion of reviewer 2.

**Figure R1.1:** a) temporal correlations and b) root mean square error (RMSE) between the annual cycle of measured and simulated precipitation sums for 2006 at the nearest grid point to the station's location are shown for "Cumulus3 1-way", ERA5, CHIRPS and IMERG. The box and whisker plots show the values in relation to the stations included in the 1 km spatial resolution domain of the 27km\_D04 setup. The whiskers extend to the value that is no more than 1.5 times the inter-quartile range away from the box. The values outside this range are defined as outliers and are plotted with dots. Pattern correlation of monthly precipitation sums between c) weather station data and the gridded data and d) between CHIRPS and the gridded data (interpolated onto the CHIRPS grid). The gridded data include ERA5, IMERG, CHIRPS and the WRF simulation "Cumulus3 1-way" for the year 2006. The labelling of the symbols is given in the table left of the two panels, along with the number of months (# months) in which the nesting option obtains correlation patterns above the reference value of 0.50 (a moderate correlation used to visually evaluate the performance of nesting options).

---

## Author Response (AR2)

**Point-by-point Reply to Referee 2**

*I would like to thank the authors for their careful and detailed responses to all of the reviewer comments, as well as for performing numerous additional sensitivity simulations to evaluate the robustness of the presented results and conclusions. The authors' revisions and clarifications have greatly improved the manuscript and the revised figures provide the relevant information in a succinct and compact way. I support its publication in GMD subject to the consideration of the small comments below in the final version.*

1. *I was curious what the difference between the 'South America' and 'Cumulus3' configurations for the 5km_D02 simulations was, because they appear identical from Table 1. A diff of the provided namelists indicates that they differ in the deep convection parameterization, which was turned on for the 5-km domain in the 'South America' configuration. I suggest that the authors check that the sample namelist is correct or amend the description of which domains employed a cumulus parameterization at line 153 in the revised manuscript/Table 1.*

Thank you very much for pointing this out. There was indeed an error in two of the 5km_D02 simulations ("Europe" and "South America"), as they included a cumulus parameterization in the outermost domain. This was a mistake and we have rerun the "Europe" simulation for this setup, now without any cumulus parameterization as describe in the manuscript. As the "South America" and "Cumulus3" are the same for the 5km_D02 setting, we have deleted the "South America" setting from the Figures (3, 4, and 5), as this is then consistent with the simulation for the "No Cumulus" setting (because it is the same as "Cumulus 3 – 1way"). In Table 1 and in the method part we indicate that the two simulations are identical and hence not shown for "South America".

2. *Figure R2.1 The results of the version testing are interesting and, in my opinion, relevant for readers interested in using the presented work to inform their own simulations. I suggest adding one sentence to the methods indicating that, compiler differences aside, using the more recent version of WRF essentially degrades the performance.*

We agree that this point is interesting for the reader and we have added some lines in the manuscript that highlight this fact. Nevertheless, we do not agree with the word "degrade", as the new version is better in some months and worse in others. We have slightly weakened this statement to:

"In addition, a simulation with the latest version of the model (V4.2.1) was run. However, it showed that the included improvements are not enough to reduce the RMSE and to improve the temporal correlation against the weather station data compared to the other sets of experiments. It further indicates that model versions and compilers can impact the simulations performed with WRF. Consequently, it has not been included in the analysis presented here."

3. *Figure S1 & Line 410: "Nevertheless, it must be noted that peak precipitation rates on a sub-daily basis are captured much more realistic with respect to IMERG, compared to the other parameterizations options, but only in domains where the others use a cumulus parameterization (see supplementary Fig. S1)."*

*It is not clear to me why the authors have included the right column (9-km/D2 in 27km_D4 set up) since the stated goal is to evaluate the impact of different configurations on the kilometer-scale solution. In addition, I don't see the result that the mean diurnal cycle is "much more realistic" in 'No Cumulus.' Although the timing of the afternoon peak is later (sometimes too late), the magnitude is very poorly captured. Can the authors clarify further the added value of showing and emphasizing these data?*

We think that this is an interesting result for the reader. The topic on the proper scale to turn off the cumulus parameterization and on the added value of convection resolved model simulations is currently strongly debated. This is why we would like to show also D02 in the Supplementary Material, as it shows resolutions used for example by CORDEX.

Nevertheless, we have adapted the statement and have pointed to the fact that mainly the timing is improved and not necessarily the precipitation amounts.